# Characterization of a new CCCTC-binding factor binding site as a dual regulator of Epstein-Barr virus latent infection

**Sun Hee Lee[1,©], Kyoung-Dong Kim[2,©], Miyeon Cho[1], Sora Huh[1], Seong Ho An[2], Donghyun Seo[1], Kyuhyun Kang[1], Minhee Lee[1], Hideki Tanizawa[3], Inuk Jung[4], Hyosun Cho[5]\*, Hyojeung Kang[1]\***

**1** College of Pharmacy, Vessel-Organ Interaction Research Center, Research Institute of Pharmaceutical Science, Kyungpook National University, Daegu, Korea, **2** Department of Systems Biotechnology, Chung-Ang University, Anseong, Korea, **3** Institute of Molecular Biology, University of Oregon, Eugene, Oregon, United States of America, **4** Department of Computer Science and Engineering, Kyungpook National University, Daegu, Korea, **5** Duksung Innovative Drug Center, College of Pharmacy, Duksung Women's University, Seoul, Korea

☯ These authors contributed equally to this work.
\* hyosun1102@duksung.ac.kr (HC); hkang72@knu.ac.kr (HK)

**Data Availability Statement:** All relevant data are within the manuscript and its Supporting information files.

## Abstract

Distinct viral gene expression characterizes Epstein-Barr virus (EBV) infection in EBV-producing marmoset B-cell (B95-8) and EBV-associated gastric carcinoma (SNU719) cell lines. CCCTC-binding factor (CTCF) is a structural chromatin factor that coordinates chromatin interactions in the EBV genome. Chromatin immunoprecipitation followed by sequencing against CTCF revealed 16 CTCF binding sites in the B95-8 and SNU719 EBV genomes. The biological function of one CTCF binding site (S13 locus) located on the *Bam*HI A right transcript (BART) miRNA promoter was elucidated experimentally. Microscale thermophoresis assay showed that CTCF binds more readily to the stable form than the mutant form of the S13 locus. EBV BART miRNA clusters encode 22 miRNAs, whose roles are implicated in EBV-related cancer pathogenesis. The B95-8 EBV genome lacks a 11.8-kb *Eco*RI C fragment, whereas the SNU719 EBV genome is full-length. ChIP-PCR assay revealed that CTCF, RNA polymerase II, H3K4me3 histone, and H3K9me3 histone were more enriched at S13 and S16 (167-kb) loci in B95-8 than in the SNU719 EBV genome. 4C-Seq and 3C-PCR assays using B95-8 and SNU719 cells showed that the S13 locus was associated with overall EBV genomic loci including 3-kb and 167-kb region in both EBV genomes. We generated mutations in the S13 locus in bacmids with or without the 11.8-kb BART transcript unit (BART(+/-)). The S13 mutation upregulated BART miRNA expression, weakened EBV latency, and reduced EBV infectivity in the presence of *Eco*RI C fragment. Another 3C-PCR assay using four types of BART(+/-)·S13(wild-type(Wt)/mutant (Mt)) HEK293-EBV cells revealed that the S13 mutation decreased DNA associations between the 167-kb region and 3-kb in the EBV genome. Based on these results, CTCF bound to the S13 locus along with the 11.8-kb *Eco*RI C fragment is suggested to form an EBV 3-dimensional DNA loop for coordinated EBV BART miRNA expression and infectivity.

**Funding:** This work was supported by 1) grants from the National Research Foundation of Korea (2018R1D1A3B07045094 (HK), 2019R1I1A3A01059629 (HK), 2022R1C1C2004274 (SHL), 2022R1I1A2066092 (HK)), 2) a grant from the Priority Research Centers Program through the National Research Foundation funded by the Korean Ministry of Education, Science, and Technology (2016R1A6A1A03007648 (HC)), 3) grants from the National Research Foundation of Korea grant funded by the Korean Government (MSIT) (2019R1F1A1061826 (KDK), 2020R1A5A2017323 (HK)), and 4) a grant from the 4TH BK21 project (Educational Research Group for Platform development of management of emerging infectious disease) funded by the Korean ministry of education (5199990614732 (HK)). The funders had no role in study design, data collection and analysis, decision to publish, or preparation of the manuscript.

**Competing interests:** The authors have declared that no competing interests exist.

## Author summary

A CCCTC-binding factor binding site (S13 locus) on the *Bam*HI A right transcript miRNA promoter was identified by Chromatin immunoprecipitation followed by sequencing and MST assays. Next, we investigated the functional roles of the S13 locus using reverse genetics followed by 4C-Seq and 3C-PCR assays. We found that in the presence of the 11.8-kb *Eco*RI C fragment, the S13 locus is associated with overall EBV genomic loci including 3-kb (S1) and 167-kb (S16). Reverse genetic studies demonstrated that the S13 locus plays an inhibitory role in EBV BART miRNA expression and lytic reactivation in the presence of the *EcoR*I C fragment. Thus, the S13 locus is suggested to form a cluster of DNA loops with the OriP (S1) and LMP1/2 (S16) loci to coordinate the EBV life cycle.

## Introduction

Epstein-Barr virus (EBV) is a member of the human gamma herpesvirus family that establishes lifelong latent infections in most host populations [1]. EBV latent infection is associated with lymphomas such as Burkitt's lymphoma (BL) and Hodgkin's lymphoma (HL), and epithelial neoplasms such as nasopharyngeal carcinoma (NPC, lymphoepithelioma) and gastric carcinoma (GC) [2, 3]. During the latency phase, the EBV genome exists as multicopy episomes that expresses only a few viral genes called latent genes [4]. Latent phase can be changed into the lytic phase depending on the developmental stage, environmental signals, and pharmacological manipulation [5, 6]. Approximately 10% GC has been diagnosed as EBV-associated gastric carcinoma (EBVaGC), with more than 70,000 cases reported worldwide per year [7–9]. EBVaGC is a lymphoepithelioma-like carcinoma, defined as an undifferentiated carcinoma with lymphocytic infiltration, and is histologically similar to NPC [2, 7]. EBV of EBVaGC maintains the type I latency phase and expresses very few EBV latent genes, such as *EBNA1*, *EBER*, BARTs, and sometimes *LMP2A*. These genes have been implicated in EBVaGC oncogenesis [10].

The EBV genome contains two miRNA clusters encoded by *Bam*HI fragment H rightward open reading frame 1 (BHRF1) and *Bam*HI A right transcripts (BARTs) [11–14]. EBV BARTs is a complex miRNA cluster of highly spliced transcripts, initially reported in NPC strains [15, 16]. Some lymphotropic EBV strains, such as B95-8, have a deletion of *Eco*RI C fragment (EBV genome (EBV) 139724–151554) overlapping with BART miRNA clusters, whereas EBV strains derived from GC, such as GC1 and YCCEL1, contain the full BART region [17, 18]. BART miRNAs are substantially expressed in EBV-infected epithelial cells such as NPC and EBVaGC [19–21]. BART miRNAs are highly implicated in EBV-mediated epithelial malignancies but are sometimes dispensable in EBV-mediated lymphoma. Thus, their function in the EBV life cycle has only been partially elucidated [22].

The maintenance of chromatin structure depends largely on cellular mechanisms that regulate chromatin interactions, exemplified by interactions between the enhancers and promoters [23–26]. CCCTC-binding factor, also referred to as CTCF, is a transcription factor that contains a DNA-binding domain and 11 zinc fingers. CTCF is also involved in other functions such as epigenetic insulators, gene boundary factors, and is involved in DNA-looping [27–29]. In particular, CTCF is highly associated with regulating long-range chromatin interactions via chromatin loop organization [30]. Cohesins composed of SMC1, SMC3, and non-SMC components, including RAD21, SA1, and SA3, assist in the CTCF-mediated stabilization of the

EBV genome structure [31–33]. Cohesins bind to multiple regulatory regions of EBV genes and along with CTCF, they are involved in maintaining the EBV genome structure to regulate EBV gene expression [34–36].

We identified 16 CTCF binding sites in EBV genome using ChIP-Seq analysis. Among them, the biological function of one CTCF binding site (S13 locus) located on the BART promoter had not yet been defined. The S13 locus exists in EBV genome regardless of the 11.8-kb *Eco*RI C fragment. Here, we tested the hypothesis that the S13 locus is important for the 3D conformation of the EBV genome and is involved in regulating BART miRNA expression. We found that the S13 locus contributes to BART miRNA expression as well as EBV 3D genome conformation. Thus, the S13 locus is likely to regulate EBV oncogenesis and latency.

## Results

### Distribution of CTCF binding site in EBV genome

The EBV-producing marmoset B-cell line B95-8 contains the B95-8 EBV genome, whereas EBVaGC cell line, SNU719 cells, carries GC1 (SNU719) EBV genome. The B95-8 EBV genome is 160-kb in length and exhibits a type-III latency program [37]. *Eco*RI C fragment (EBV 139724~151554, 11.8-kb), is conserved in 883 EBV strains but is lost in the B95-8 EBV genome [38]. This deleted region in the B95-8 EBV genome includes two clusters of BART miRNAs. The SNU719 EBV genome is 170-kb in length and exhibits the type-I latency program. Compared to the wild-type EBV genome (NC_007605), the SNU719 EBV genome harbors the same mutations in *EBNA1* and *LMP1* as the GD1 EBV genome [39]. Both B95-8 and SNU719 cells are good *in vitro* models for studying comparative EBV genomics. Due to these features, we performed a CTCF chromatin immunoprecipitation followed by sequencing (ChIP-Seq) assay using B95-8 and SNU719 cells to identify CTCF binding sites in EBV genome (Fig 1A). The sequencing reads were mapped to the EBV reference genome (NC_007605) and visualized using the Integrative Genomics Viewer [40].

ChIP-Seq assay using B95-8 cells produced approximately $3.9 \times 10^6$ CTCF ChIP reads and $2.7 \times 10^6$ IgG reads, while ChIP-Seq assay using SNU719 cells produced approximately $4.4 \times 10^6$ CTCF ChIP reads and $3.2 \times 10^6$ IgG reads. Enrichment of CTCF ChIP products relative to IgG ChIP products was calculated and visualized as 95% confidence interval lower bound. Peaks indicating CTCF enrichment were identified at more than 16 places based on read depth in both cells (S1 Table). High-confidence peaks such as S1, S3, S4, S5, S8, S11, S13, S14 and S16 were clearly observed in both cells. However, low-confidence peaks such S2, S6, S7, S9, S12, and S15 were minutely distinguished in B95-8 cells but not separated in SNU719 cells. In addition to these sites, the sequence of S4 locus (EBV 40932–40981) was identical to the S14 locus (EBV 143851–143900). Many CTCF binding sites were adjacent to the 5' and 3' ends of EBV genes, suggesting that they are located in the regions regulating transcription of EBV genes.

Multiple Em for motif elicitation (MEME) was used to generate a sequence logo for the CTCF binding sites identified by the CTCF ChIP-Seq assay (Fig 1B). The MEME P-values for S3, S1, S16, S5, S11, S8, S4/14, and S13 ranged between $3.86e^{-8}$ to $3.91e^{-5}$. Compared with the logos of CTCF binding sites generated in previous studies [41], the logo in our study was similar to the LowOc logo or the MedOc logo in the three classes of CTCF binding sites.

### Binding of CTCF on the BART promoter region

The functional roles of CTCF binding sites in the EBV genome have been identified in EBV-associated lymphoid cell lines, such as Mutu I, Mutu-LCL, and Raji. CTCF binding sites, such as S1, S2, S5, and S16 in Fig 1A, have been characterized in previous studies and implicated in

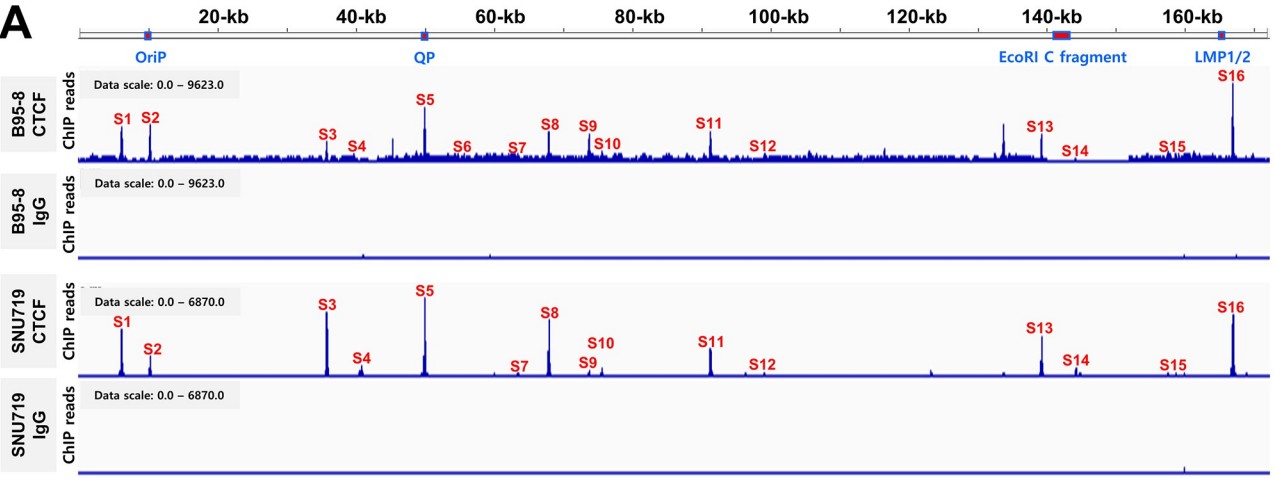

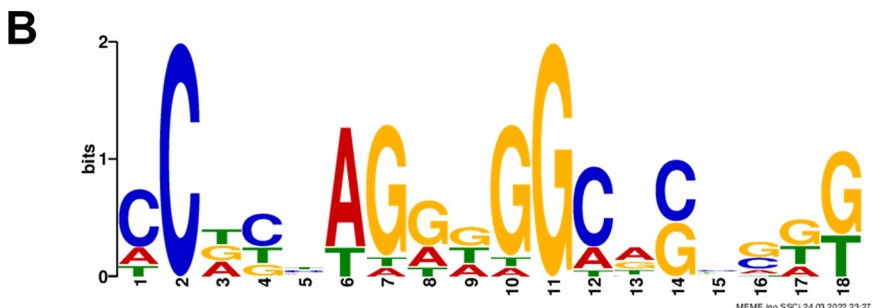

**Fig 1. Identification of CTCF binding site on EBV genome in B95-8 and SUN719 cell line. A)** CTCF enrichment in EBV genomes of B95-8 and SNU719 cells as determined by CTCF ChIP-seq assays, with IgG ChIP-seq assays performed as negative controls. The mac2 tool was used for peak calling. Next, bigwig files of the ChIP-seq data were applied to Integrative Genomics Viewer (IGV) with EBV reference genome (NC_007605). Resultant peaks were selected and marked as S1 to S16 to indicate CTCF enrichment. **B)** CTCF Logo. Sequences of S1 to S16 were harvested using IGV and applied to MEME using which a LoGo of CTCF binding sites was drawn.

DNA-looping [42–44]. However, the S13 locus (EBV 138901~138994) located near the EBV BART promoter (S1 Table, EBV 137821~138420) has not been previously characterized.

To understand the function of the S13 locus in CTCF binding, we first performed Coomassie blue staining analysis to determine the quality of CTCF protein purified from Sf9 cells using the baculovirus CTCF expression system (Fig 2A). CTCF protein was of good quality and suitable for subsequent experiments. We then conducted a microscale thermophoresis (MST) assay to confirm CTCF binding to the S13 locus as identified by a previous CTCF ChIP-Seq assay. MST assay is a unique tool to quantify the binding affinity between protein and DNA [45]. The principle of MST analysis is thermophoresis, which involves the movement of fluorescent molecules through a temperature gradient depending on the concentration of potential binding partners [45]. The wild-type (Wt) S13 oligo used in the MST assay was designed using the S13 locus (EBV 138946–138994), and the mutant (Mt) S13 oligo was constructed by introducing site-directed mutations (EBV 138963–138978) to the Wt S13 oligo (Fig 2B). We labelled wild-type or mutant oligonucleotides with a carboxyfluorescein (FAM) and used it at a constant concentration and increased the concentrations of Sf9 lysates and purified CTCF proteins.

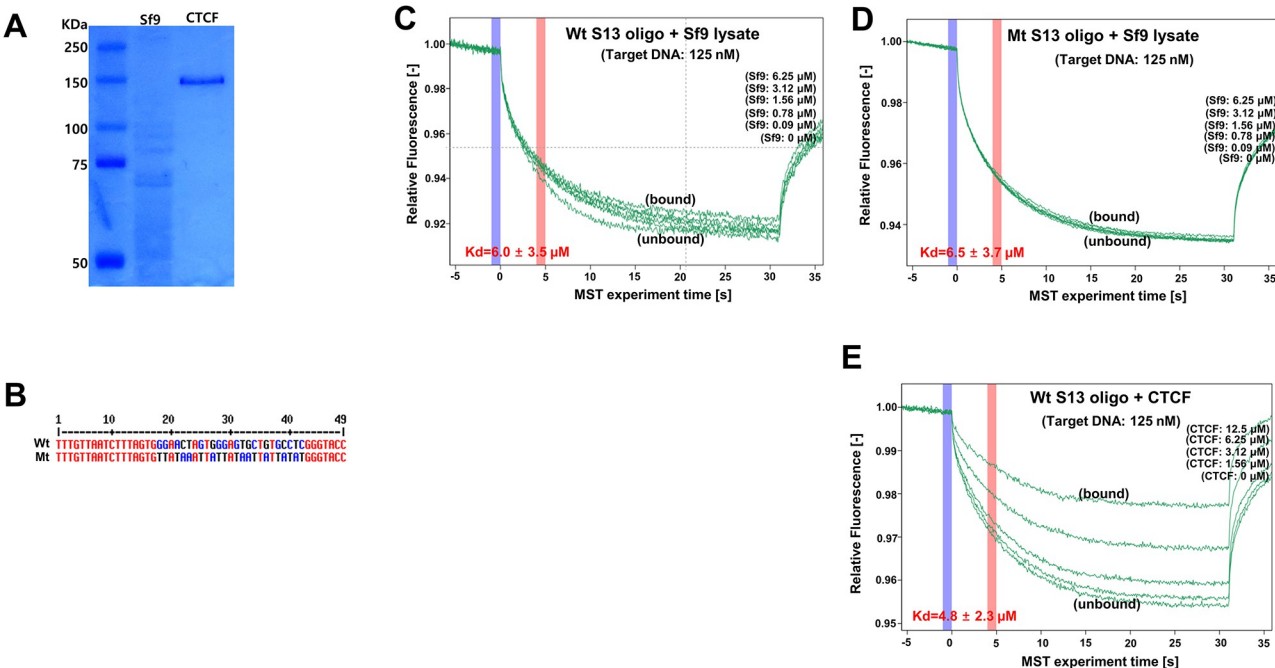

**Fig 2. Detection of CTCF-DNA interaction using the Monolith NT.115. A)** Quality confirmation of purified CTCF protein by Coomassie blue staining. CTCF was purified using Sf9 baculovirus protein expression system. **B)** S13 5' FAM labeled 49-mer primer whose sequence was mutated (Mt S13 primer sets) or not (Wt S13 primer sets). Wt S13 locus used in this study was designed from EBV 138946 ~ 138994 (NC_007605), and Mt S13 locus was constructed by introducing site-directed mutations into EBV 138963 ~ 138978. **C-E)** CTCF-DNA interaction tested by Microscale thermophoresis (MST) assay. Wild-type (Wt) and mutant (Mt) S13 5' FAM labeled 49-mer primer sets were paired to convert double DNA fragments. MST assays for determining CTCF binding to Wt and Mt S13 DNA fragments were conducted using 125 nM DNA fragments and a series of concentration of purified CTCF proteins (0 μM, 1.56 μM, 3.12 μM, 6.25 μM, and 12.5 μM); MST assay for binding of Wt S13 oligo to Sf9 lysate (C), MST assay for binding of Mt S13 oligo to Sf9 lysate (D), and MST assay for binding of Wt S13 oligo to purified CTCF protein (E). Blue vertical and pink lines indicate time of initial fluorescence (no molecule flow) and time of thermophoresis/backdiffusion (molecule flow), respectively.

We then conducted an MST assay to confirm CTCF binding to the Wt S13 oligo and Mt S13 oligo (Fig 2C–2E). We observed strong bindings of purified CTCF proteins (Fig 2E, Kd value = 4.8 ± 2.3 μM) to the Wt S13 oligo. In contrast, binding of Sf9 lysates to Wt or Mt S13 oligo was significantly compromised. Actually, Sf9 cellular lysates showed no strong selectivity for binding to the Wt S13 oligo (Fig 2C, Kd value = 6.0 ± 3.5 μM) or Mt S13 oligo (Fig 2D, Kd value = 6.5 ± 3.7 μM). These data indicate that the Wt S13 locus located on the BART promoter is a good target site for CTCF binding.

## Investigation of epigenetic factors acting on the BART promoter region

The S13 locus is located near the BART promoter region and within the intron of *RPMS1* gene. Given its location, we speculated that the S13 locus functions as a transcriptional regulator and/or DNA loop maker, thereby mediating transcriptional regulation via chromatin interactions in the EBV genome. Furthermore, we sought to investigate the epigenetic factors that may influence the transcriptional regulation function of the S13 locus in the EBV genome. For this purpose, ChIP-qPCR assays were performed for CTCF, RNA polymerase II (RNAP II), H3K4me3, and H3K9me3 using B95-8 and SNU719 cells (Fig 3). A series of primers were used to detect ChIP enrichment at the target loci (S2 Table). We found that CTCF was more enriched at the S16 locus in B95-8 cells than in SNU719 cells, although it was similarly enriched at the S13 locus in both cells (Fig 3A). In contrast, RNAPII was more enriched at the

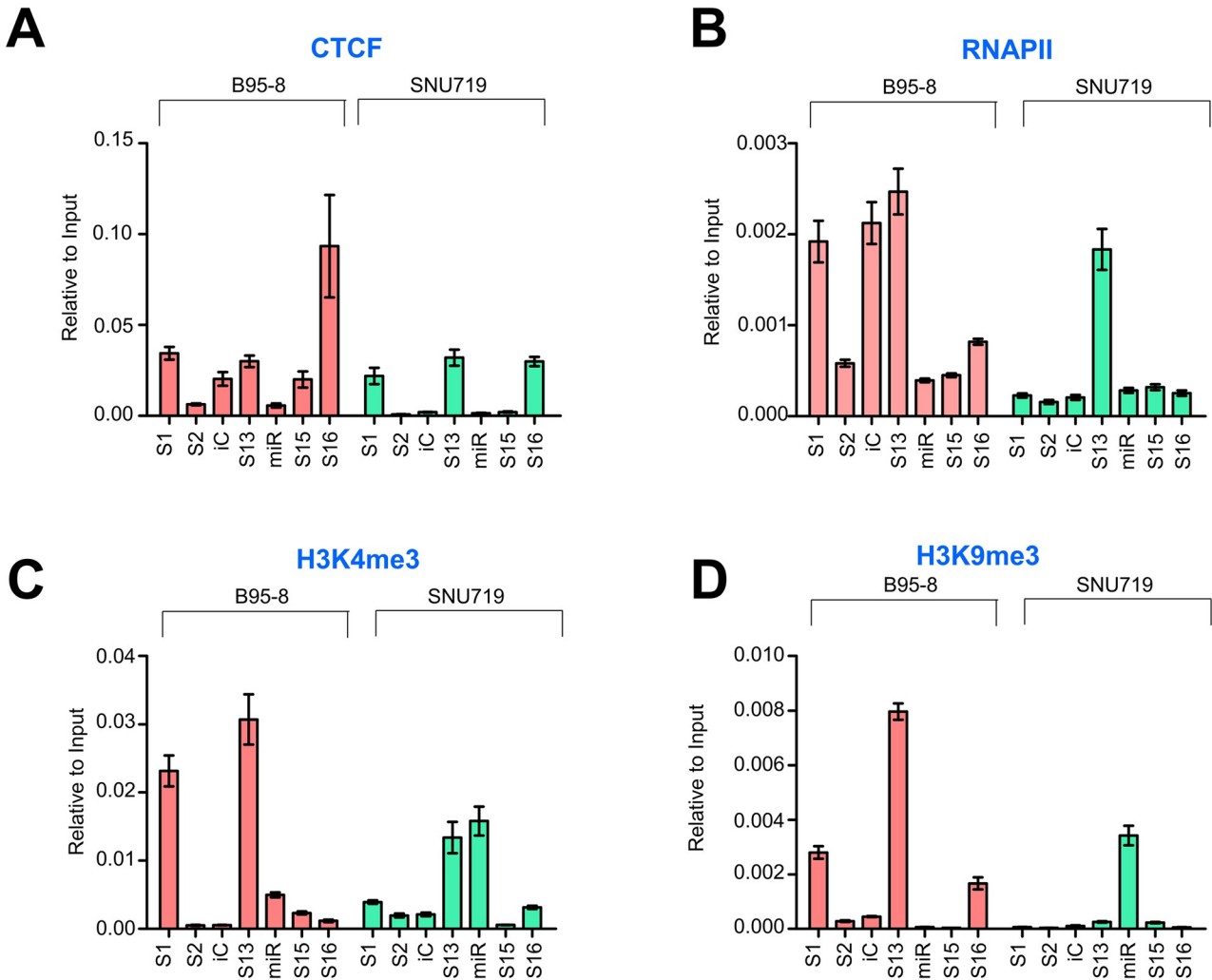

**Fig 3. Analysis of epigenetic factors nearby CTCF binding sites.** ChIP-qPCR assay was conducted to identify epigenetic factors nearby EBV CTCF binding sites between B95-8 and SNU719 cells; **A)** enrichment of CTCF at the suggested CTCF binding sites, **B)** enrichments of RNAP II at the suggested CTCF binding sites, **C, D)** enrichments of histones such as H3K4me3 and H3K9me3, at the suggested CTCF binding sites. Genomic locations of S1, S2, iC, S13, and S16 are shown in S2 Table. iC (EBV 38173 ~ 38272) locus was used as a negative control because none of the CTCF bound to this site in B95-8 and SUN719 cells. In all panels, experiments were independently repeated two times, and data are represented as mean ± SD.

S16 and S13 loci in B95-8 cells than SNU719 cells (Fig 3B). Furthermore, H3K4me3 histone was more highly enriched at the S16 and S1 loci in B95-8 cells than in the SNU719 cells (Fig 3C), while H3K9me3 histone was more enriched at the S1 and S13 loci in B95-8 cells than in SNU719 cells (Fig 3D). In general, CTCF and RNAP II were recruited more at the S13 and S16 loci in the B95-8 EBV genome than in the SNU719 EBV genome. Considering that the 11.8-kb *Eco*RI C fragment (EBV 139724~151554) exists only in the SNU719 EBV genome, it may play an unfavorable role in recruiting CTCF and RNAP II in the SNU719 EBV genome. Overall, these results suggest that the S13 locus might be involved in expressing BART miRNAs by coordinating the enrichment of CTCF, RNA polymerase, and active histone markers.

## Comparison of BART miRNA expression in EBV genome

A previous ChIP-Seq assay revealed that the S13 locus is located upstream of BART3, the first miRNA in BART cluster 1 (S1 Table). Moreover, we observed a dense amount of RNAPII

enriched at the S13 locus, along with H3K4me3 marker. These results led us to speculate that the S13 locus may be implicated in the expression of BART miRNAs. Therefore, we investigated expression patterns of BART miRNAs in B95-8 and SNU719 cells. Before determining expression patterns of BART miRNAs and EBV genes, we induced EBV lytic reactivation by treating these B95-8 and SNU719 cells with TPA (20 ng/mL) and NaB (3 mM) for 48 h and compared also the induced cells with the uninduced cells. Five miRNAs were selected from two BART miRNA clusters, miR-BART1-5p, miR-BART15, miR-BART6-5p, miR-BART11-5p, and miR-BART2-5p (Fig 4A). We found that miR-BART1-5p, miR-BART15, and miR-BART2-5p were more highly expressed in B95-8 cells than SNU719 cells. Furthermore, miR-

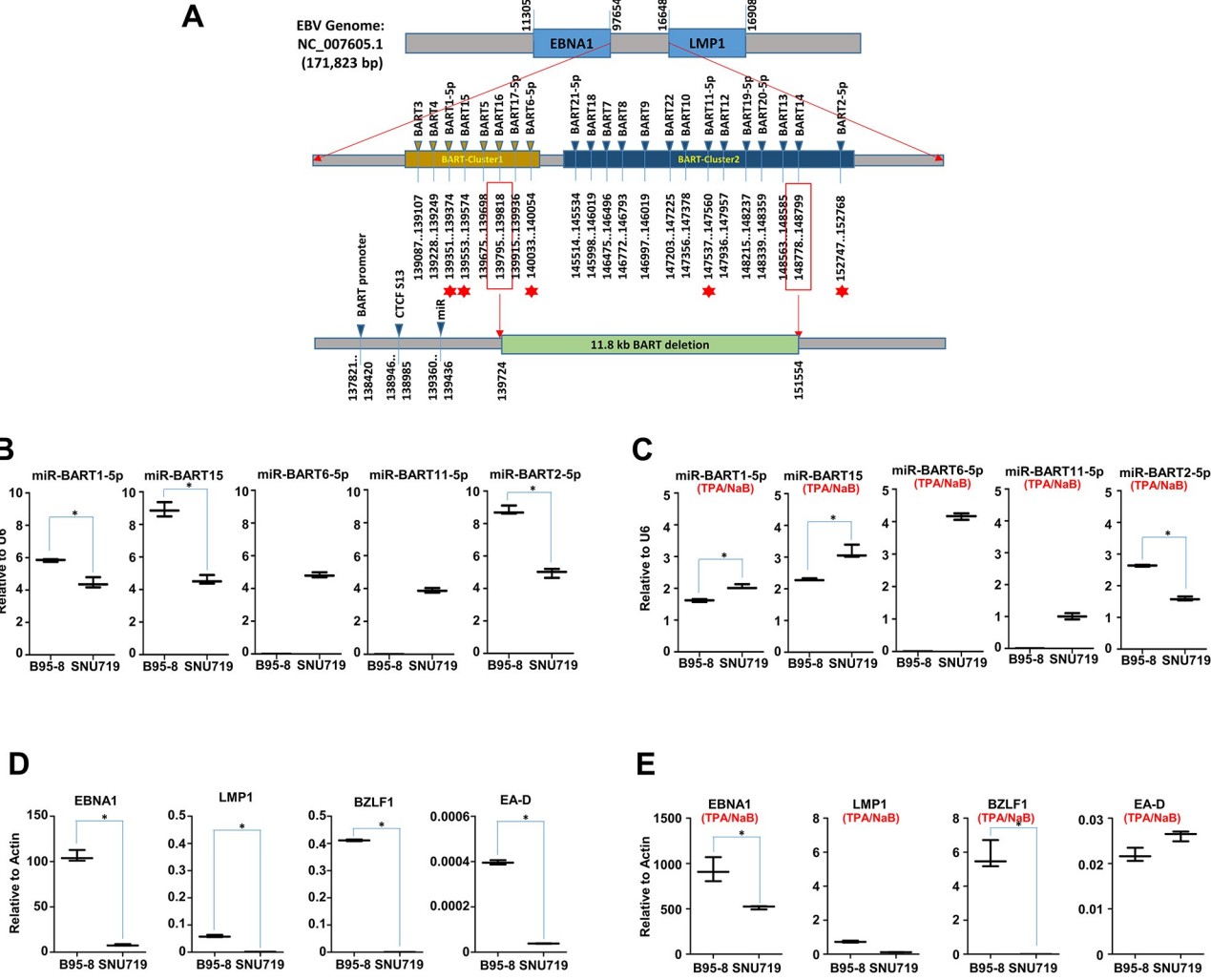

**Fig 4. Analysis of BART miRNAs. A)** Simple map of BART miRNAs in EBV BART miRNA cluster composed of BART cluster 1 and BART cluster 2. An 11.8 kb BART deletion was located downstream of CTCF binding site named as S13. Five BART miRNAs (BART1-5p, BART15, BART6-5p, BART11-5p, and BART2-5p) were selected to investigate expression patterns of BART miRNAs in B95-8 and SNU719 cells. **B)** Comparison of expression patterns of the selected five BART miRNAs between uninduced B95-8 and SNU719 cells. Due to the absence of miR BART6-5p and miR BART11-5p in B95-8 cells, their expression was not detectable in B95-8 cells. Expression of each miRNA was normalized with that of U6 expression. **C)** Comparison of expression patterns of the selected five BART miRNAs between induced B95-8 and SNU719 cells. **D)** Comparison of expression patterns of the selected EBV genes between uninduced B95-8 and SNU719 cells. Expression of each EBV gene was normalized with that of actin expression. **E)** Comparison of expression patterns of the selected EBV genes between induced B95-8 and SNU719 cells. Statistical analysis were performed using paired t test. In all panels, experiments were independently repeated two times, and data are represented as mean ± SD.

BART6-5p and miR-BART11-5p were expressed only in SNU719 cells due to the deleted 11.8-kb *Eco*RI C fragment in B95-8 cells (Fig 4B). In induced conditions, miR-BART1-5p and miR-BART15 were more upregulated in SNU719 cells than B95-8 cells (Fig 4C). Since miR--BART1-5p and miR-BART15 were located close to the S13 locus, we could not exclude the possibility that the S13 locus might be associated with differential expression of miR-BART1-5p and miR-BART15 between two cells. In similar perspective, transcripts of EBV genes such as *EBNA1*, *LMP1*, *BZLF1*, and *EA-D* were also more abundant in B95-8 cells than SNU719 cells (Fig 4D). However, *EA-D* gene in induced conditions were slightly transcribed in SNU719 cells than B95-8 cells (Fig 4E). Taken together, these results indicated that higher expression of miRNAs and EBV genes in the B95-8 EBV genome is related to increased recruitment of RNA polymerase at S13 and deletion of the 11.8-kb *Eco*RI C fragment. Hence, differential expression patterns between the B95-8 and SNU719 EBV genomes might provide insight into the biological role of the S13 locus in the EBV life cycle.

## Investigation of chromatin interaction mediated by S13 locus in EBV genome

To examine whether the S13 locus is involved in the formation of DNA loops, a 4C-Seq assay was conducted using B95-8 and SNU719 cells. Nuclei isolated from both cell types were permeabilized, DNA-digested, and ligated to generate 4C-Seq products. Amplicons of 4C-Seq products obtained using several different primers were sequenced, followed by the alignment of the resultant sequencing reads to the EBV genome (S1A Fig). The genomic association with the S13 locus was first investigated. The S13 primer set (EBV 138504–138528, EBV 138970–138987) was used to amplify DNA fragments associated with the S13 locus (S3 Table). Filtrated reads were adjusted to 3,000 reads for the analysis of each cell (S5 Table) and same DNA-DNA association should appear at least twice in the 4C-seq analysis. Interestingly, the B95-8 S13 locus were associated with a wide range of EBV genomic loci (Fig 5A). The S13 locus in the B95-8 EBV genome was associated uniquely the 1~5-kb region (near S1) and 45~50-kb region (near S5), compared the locus in the SNU719 EBV genome (Fig 5A). However, the S13 locus in the SNU719 EBV genome was linked uniquely the 148~153-kb region (near BART), compared the locus in the SNU719 EBV genome (Fig 5A).

Beside the S13 locus, we investigated EBV genomic associations of other loci in similar manner except adjusting filtered reads and limiting DNA-DNA associations. Total filtrated reads from all tested viewpoints (BART-1, BART-2, LMP1/2, Qp, OriP, and FR) were adjusted to 7,000,000 for the analysis of each cell (S4 Table). All DNA-DNA associations (including at least one association) were determined in parallel to compare between EBV genomes in B95-8 and SNU719 cells (S4 Table). The BART-1 primer set (BART-1: EBV 147470–147490) was used to amplify DNA fragments associated with the BART locus in the 4C-Seq assay (S3 Table). The BART1 viewpoint does not exist in B95-8 EBV genome, hence found no reads in the B95-8 4C-seq assay (Fig 5B). The BART locus in the SNU719 EBV genome was associated selectively with the 45~50-kb region (near S5), 80~85-kb region (near S11), 120~125-kb region, and 155~160-kb region (near S16) (Fig 5B). The LMP primer set (EBV 166661–166680 and EBV 166501–166524) was used to amplify DNA fragments associated with the LMP1/2 locus (near S16). LMP1/2 loci in B95-8 and SNU719 EBV genomes were associated selectively with the 45~55-kb region (near S5), 70~75-kb region (near S8), 95~100-kb region (near S11), 110~115-kb region, and 125~130-kb region (Fig 5C). The B95-8 LMP1/2 locus showed slightly weaker interactions with any region in the 5' terminus of the EBV genome than the SNU719 LMP1/2 locus (Fig 5B). The Qp primer set (EBV 50278–50294, EBV 51560–51579) was used to amplify DNA fragments associated with the Qp locus (near S5).

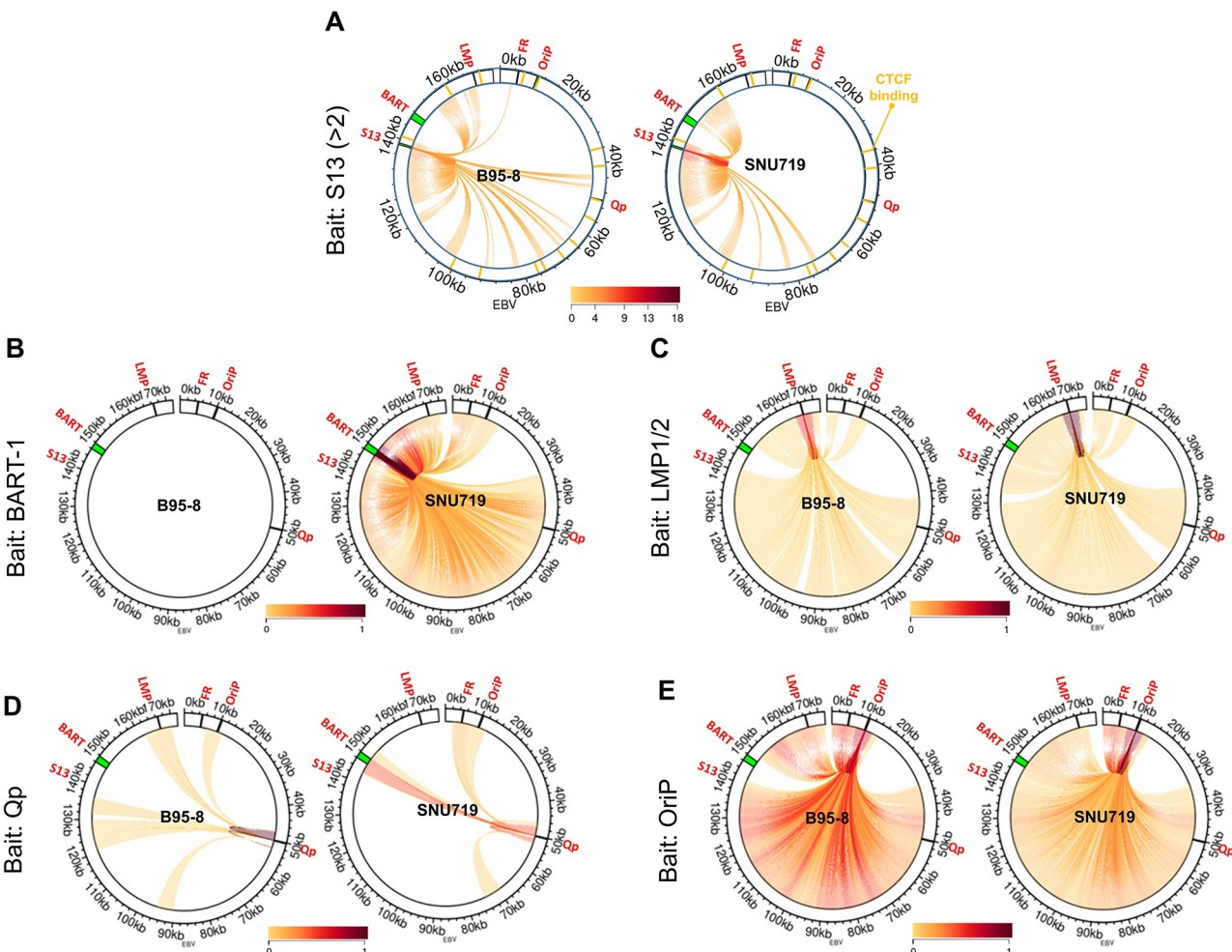

**Fig 5. 4C-plots on EBV genome.** Significant genomic associations of target viewpoint regions in B95-8 and SNU719 EBV genomes identified by 4C-Seq assays. **A)** To compare EBV genomic associations with the S13 locus, we adjusted the 4C-seq read numbers from S13 viewpoint primer set to 3,000 in the 4C-seq analysis of B95-8 and SNU719 cells (S5 Table). **B-E)** To compare EBV genomic associations with other loci such as BART-1, LMP1/2, Qp, and OriP, total read numbers from all target viewpoints were adjusted to 7,000,000 in each cell (S4 Table). 4C-seq assays revealed all interactions of BART-1 **(B)**, LMP1/2 **(C)**, Qp **(D)**, and OriP **(E)** regions in B95-8 and SNU719 EBV genomes. Locations of viewpoint primer sets were posted in S3 Table; S13 (EBV 138504 ~ 138987), BART-1 (EBV 147357 ~ 147490), LMP1/2 (EBV 166480 ~ 166661), Qp (EBV 50278 ~ 51579), OriP (EBV 9918 ~ 10302) and FR (EBV 4731 ~ 4961).

The B95-8 Qp locus was uniquely associated with only the B95-8 LMP1/2 locus and additionally linked to the OriP locus, 90~95-kb region, and 125~130-kb region. The SNU719 Qp locus highly interacted with several SNU719 loci, such as the BART, FR, OriP, 75–80-kb region, and 115–125-kb region (Fig 5D). The OriP primer set (EBV 9918–9940, EBV 10281–10302) amplified DNA fragments associated with the OriP locus (near S2). The OriP loci in EBV genomes of B95-8 and SNU719 cells were innumerably associated with multiple loci in the EBV genomes (Fig 5E). At last, the FR primer set (EBV 4731–4751, EBV 4941–4961) amplified DNA fragments associated with the FR locus (near S1). Like the OriP locus, the FR loci in EBV genomes of B95-8 and SNU719 cells were also innumerably associated with multiple loci in the EBV genomes (S1B Fig). DNA association patterns of FR and OriP loci were similar each other because the location of two viewpoints are in close proximity. Overall, the B95-8 viewpoint loci showed slightly more interactions within EBV genome than the SNU719

viewpoint loci. In particular, B95-8 EBV genome has more active genomic association with OriP/FR and S13 loci, compared to SNU719 EBV genome.

A 3C-PCR assay was conducted to consolidate the multiple chromatin interactions identified by the 4C-Seq assay. Nuclei isolated from B95-8 and SNU719 cells were subjected to paraformaldehyde fixation, *Xho*I-digestion, T4 DNA ligase ligation, and PCR assays. The CTCF binding site (S13 locus) in the EBV BART promoter was included in an EBV DNA fragment (EBV 135936~147676) generated by digesting EBV genome with *Xho*I. The viewpoint primer sets for the 3C-PCR assay were selected from the CTCF binding sites identified by the previous ChIP-Seq assay (S5 Table). The primer (EBV 136109–136129) designed based on a 135-kb region (near the S13 locus) in the reverse direction was tested for chromatin interactions with other important loci in EBV genome (S5 Table). The B95-8 135-kb region were likely to interact with EBV genome components, such as the 3-kb region (near S1 locus), 35-kb region (near S3 locus), 65-kb region (near S11 locus), 88-kb region (near S11 locus), and 167-kb region (near S11 locus) (S2A Fig). As a control, we conducted a PCR assay with unligated *Xho*I-digested B95-8 DNA samples under the same conditions as those used for PCR with ligated *Xho*I-digested B95-8 DNA samples; no false positive 200~500 bp band was amplified from the 3C-PCR primer sets (S3A Fig). We further confirmed by sequencing the interactions of B95-8 EBV 135-kb region with the 3-kb region and 167-kb regions (S4 Fig). The SNU719 135-kb region was likely to interact with the 3-kb region, the 65-kb region, 88-kb region, and 167-kb region (S2B Fig). As a control, we conducted a PCR assay with unligated *Xho*I-digested SNU719 DNA samples under the same conditions as those used for PCR with ligated *Xho*I-digested SNU719 DNA samples; no false positive 200~500 bp band was amplified from the 3C-PCR primer sets (S3B Fig). We further confirmed by sequencing the interactions of SUN719 EBV 135-kb region with the 3-kb region and 167-kb regions (S4A and S4B Fig). These results indicated that the S13 locus is highly intermingled with several other loci in the EBV genome.

## Establishment of Mt BART(+/-)·S13⁻ HEK293-EBV cells

The B95-8 and SNU719 EBV genome sequences were compared in a previous study [46]. This comparison revealed that the 11.8-kb *Eco*RI C fragment was absent in the B95-8 EBV genome but was present in the SNU719 EBV genome. However, the loss of the 11.8-kb *Eco*RI C fragment did not remove the S13 locus from the B95-8 EBV genome, indicating that the S13 locus can be functional in both B95-8 and SNU719 EBV genomes (S5A Fig). In addition, since the S13 locus is located in the *RPMS1* intron region, the introduction of a site-directed mutation in the S13 locus did not affect *RPMS1* expression (S5B Fig).

The BART(-)·S13⁺ EBV bacmid was constructed based on the B95-8 EBV genome, in which the 11.8-kb *Eco*RI C fragment was deleted [38]. The BART(+)·S13⁺ EBV bacmid was constructed by inserting an 11.8-kb *Eco*RI C fragment into the BART(-)·S13⁺ EBV bacmid [47]. The red-recombination system produced Mt BART(+/-)·S13⁻ EBV bacmids, whose S13 loci had site-specific mutations in BART(+/-)·S13⁺ EBV bacmids. Sanger sequencing confirmed the site-directed mutation in the S13 loci of the Mt BART(+/-)·S13⁻ EBV bacmids (S5C Fig). Thereafter, four types of BART(+/-)·S13 (BART(+)·S13⁺, Mt BART(+)·S13⁻, BART(-)·S13⁺, and Mt BART(-)·S13⁻) EBV bacmids were further tested for their stabilities following *Eco*RI-digestion and we did not find any additional loss of EBV DNA using these methods (S5D Fig). Next, we transfected these four types of BART(+/−)·S13 EBV bacmids into HEK293 cells and harvested them all four types of BART(+/−)·S13 HEK293-EBV cells by selecting with hygromycin B. These cells showed no defects in their ability to express GFP even after several passages (S5E Fig).

To further investigate the EBV episome establishment in four types of BART(+/-)·S13 HEK293-EBV cells, we conducted a Southern blot assay using these cells (S5F Fig). The DNA probe was targeted to part of the EBV *BLLF1* gene (EBV 78803–79522), whose signals were sufficient to define the EBV episome in genomes of BART(+/-)·S13 HEK293-EBV cells. We quantified band intensity by densitometry of several bands of this Southern blot analysis (S7 Table). The four types of BART(+/-)·S13 HEK293-EBV cells maintained EBV episomes at comparable intensities. Based on band density, the S13 mutation slightly decreased the number of EBV episomes in Mt BART(-)·S13⁻ HEK293-EBV cells but slightly increased their numbers in Mt BART(+)·S13⁻ HEK293-EBV cells (S5F Fig).

To further confirm the S13 mutation on EBV episome establishment in four types of BART(+/-)·S13 HEK293-EBV cells, we conducted CTCF ChIP-qPCR assay using four types of BART(+/-)·S13 HEK293-EBV cells and found that site-directed mutagenesis at the S13 locus was sufficient to significantly suppress the capacity of CTCF to bind to the S13 locus (S5G Fig). We also tested if mutation in the S13 locus affects *RPMS1* gene transcription in four types of BART(+/-)·S13 HEK293-EBV cells. RT-qPCR assays showed that RPMS1 expressions in Mt BART(-)·S13⁻ and BART(+)·S13⁻ EBV genomes were reduced up to 32% and 2% in comparison with Wt BART(-)·S13⁺ and BART(-)·S13⁺ EBV genomes, respectively (S5H Fig).

## Determination of biological roles of S13 locus

The biological effects of the S13 locus on EBV latency and lytic reactivation were investigated. To this end, we measured intracellular and extracellular EBV genome copy numbers in four types of BART(+/−)·S13 HEK293-EBV cells. Before measuring EBV genome copy numbers, we induced EBV lytic reactivation by treating these HEK293-EBV cells with TPA (20 ng/mL) and NaB (3 mM) for 48 h and compared the induced HEK293-EBV cells with the uninduced HEK293-EBV cells. In uninduced or induced conditions, intracellular Mt BART(+)·S13- EBV genome copy numbers were significantly lower than those of Wt BART(+)·S13+ EBV genome (Fig 6A). Extracellular Mt BART(-)·S13- EBV genome copy numbers in uninduced or induced conditions were significantly lower than those of Wt BART(-)·S13+ EBV genome (Fig 6A). However, extracellular Mt BART(+)·S13- EBV genome copy numbers in uninduced condition were significantly higher than those of Wt BART(+)·S13+ EBV genome (Fig 6A). In comparison of induced Wt BART(-)·S13+ and Wt BART(+)·S13+ EBV genomes, extracellular and intracellular Wt BART(+)·S13+ EBV genome copy numbers were significantly lower than those of Wt BART(-)·S13+ EBV genome (S7 Fig). However, extracellular Mt BART(+)·S13- EBV genome copy numbers in uninduced condition were significantly lower than those of Mt BART(-)·S13- EBV genome (S7 Fig).

Next, we tested whether the S13 mutation regulated downstream BART miRNA expression. The S13 mutation in uninduced Mt BART(-)·S13⁻ EBV genome downregulated the expressions of miR-BART1-5p, miR-BART15, and miR-BART2-5p (Fig 6B). In contrast, the S13 mutation in uninduced Mt BART(+)·S13⁻ EBV genome significantly upregulated the expression of miR-BART1-5p, miR-BART15, miR-BART6-5p, miR-BART11-5p, and miR-BART2-5p (Fig 6B). Next, The S13 mutation in induced Mt BART(-)·S13⁻ EBV genome downregulated the expressions of miR-BART2-5p in comparison with induced Wt BART(-)·S13⁺ EBV genome (Fig 6C). Furthermore, the S13 mutation in induced Mt BART(+)·S13⁻ EBV genome significantly downregulated the expression of miR-BART1-5p, miR-BART15, miR-BART6-5p, miR-BART11-5p, and miR-BART2-5p in comparison with induced Wt BART(+)·S13⁺ EBV genome (Fig 6C). Combined with the previous results, these results indicate different regulatory roles of the S13 locus in BART miRNA expression, which is dependent on the 11.8-kb *Eco*RI C fragment. The S13 locus plays an essential factor to normally express BART miRNAs

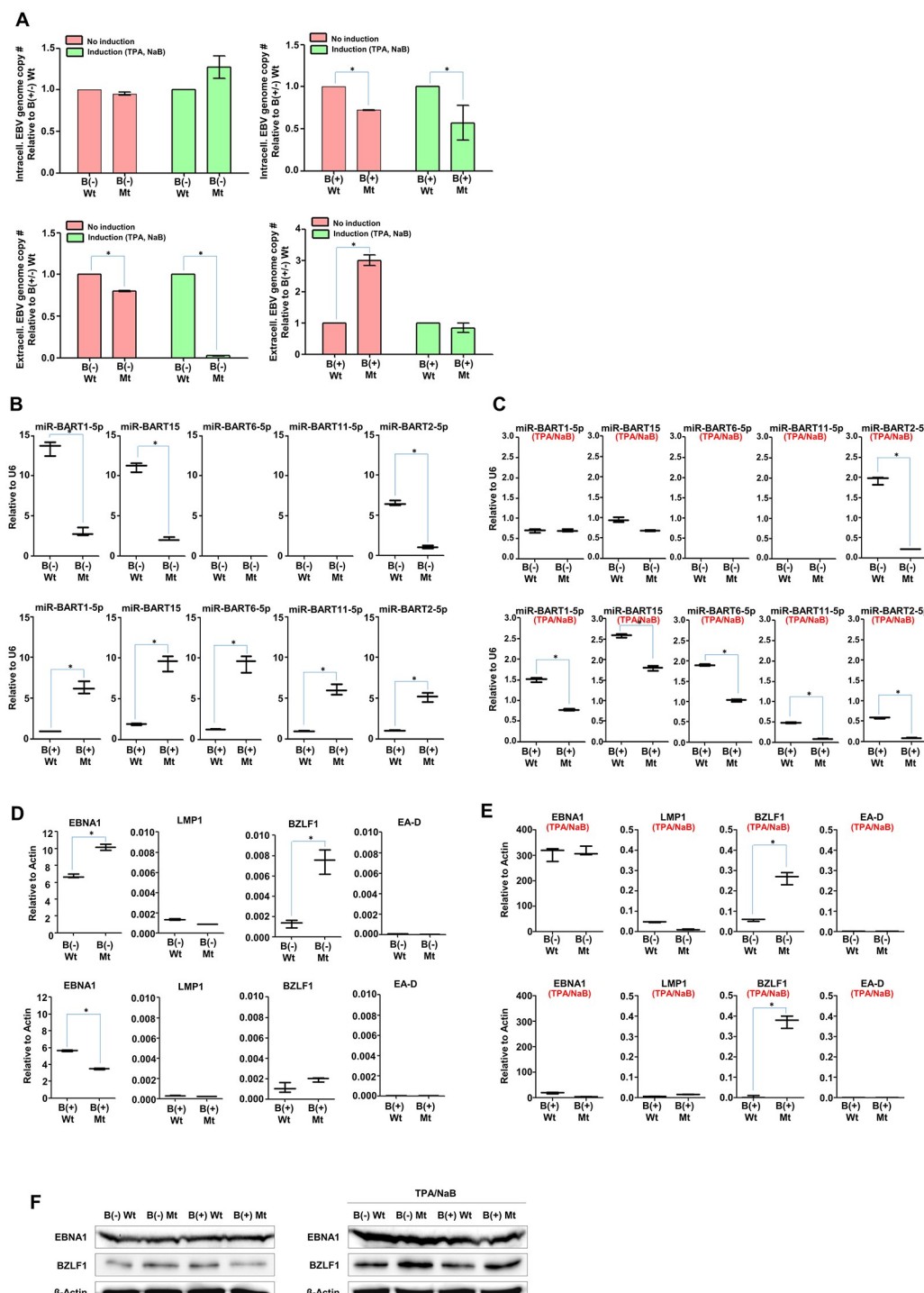

**Fig 6. Regulatory role of S13 CTCF binding site on EBV gene expression. A)** Relative quantification of relative intracellular and extracellular EBV genome copy numbers. qPCR assay with *EBNA1* primer set (EBV 96778–96979, EBV 96827–96845) was conducted to measure EBV genome copy numbers in all four types of HEK293-EBV cells which were either uninduced or induced with TPA and NaB. B(-)Wt, B(-)Mt, B(+)Wt, and B(+)Mt stand for HEK293-EBV cells of BART(-)·S13+, BART(-)·S13-, BART(+)·S13+, BART(+)·S13-, respectively. Absolute intracellular and extracellular EBV genome copy numbers were Both absolute intracellular and extracellular EBV genome copy numbers were evaluated by *EBNA1* Ct values. Relation equation between absolute DNA amounts and *EBNA1* Ct values was first defined and then applied to calculate DNA concentration (μg/μL) per length of template (bp) using a website called Copy Number Calculator of Technology NetWorks (https://www.technologynetworks.com/tn/tools/copynumbercalculator). Finally,

Both absolute intracellular and extracellular EBV genome copy numbers were calculated by relatively comparing EBV genome copy numbers of Mt BART(+/-)·S13⁻ EBVs with EBV genome copy numbers of Wt BART(+/-)·S13⁺ EBVs. **B)** Comparison of expression patterns of the selected five BART miRNAs between uninduced Wt BART(+/-)·S13⁺ and Mt BART(+/-)·S13⁻ HEK293-EBV cells. **C)** Comparison of expression patterns of the selected five BART miRNAs between induced Wt BART(+/-)·S13⁺ and Mt BART(+/-)·S13⁻ HEK293-EBV cells. 3 mM NaB and 20 μg/mL TPA were used to induce EBV lytic reactivation. **D)** Comparison of transcription patterns of EBV major genes between uninduced BART (+/-)·S13⁺ and Mt BART(+/-)·S13⁻ HEK293-EBV cells. **E)** Comparison of transcription patterns of EBV major genes between induced BART(+/-)·S13⁺ and Mt BART(+/-)·S13⁻ HEK293-EBV cells. **F)** Comparison of expression patterns of EBNA1 and BZLF1 proteins between BART(+/-)·S13⁺ and Mt BART(+/-)·S13⁻ HEK293-EBV cells in uninduced or induced conditions. In panels, experiments were independently repeated two times, and data are represented as mean ± SD. Statistical analysis were performed using paired t test.

in uninduced or induced Wt BART(-)·S13⁺ EBV genome. In contrast, the S13 locus played a suppressor in uninduced Wt BART(+)·S13⁺ EBV genome but an essential factor in induced Wt BART(+)·S13⁺ EBV genome.

Next, we tested whether the S13 mutation affected EBV gene expression. RT-qPCR assays revealed that transcripts of *EBNA1* and *BZLF1* genes were significantly more abundant in uninduced Mt BART(-)·S13⁻ EBV genome than those in uninduced Wt BART(-)·S13⁺ EBV genome (Fig 6D). In contrast, the *EBNA1* transcript was slightly less abundant in uninduced Mt BART(+)·S13⁻ EBV genome than those in uninduced Wt BART(+)·S13⁺ EBV genome (Fig 6E). Next, RT-qPCR assays revealed that the *BZLF1* transcript was significantly abundant in induced Mt BART(+/-) S13⁻ EBV genome than those in induced Wt BART(+/-)·S13⁺ EBV genome (Fig 6E). Relative high expression of *BZLF1* transcript in Mt BART(+/-)·S13⁻ EBV genome were not directly related to the increases of extracellular EBV genome copy numbers in Mt BART(+/-)·S13⁻ HEK293-EBV cells. These results suggested that the S13 locus is involved in regulating the abortive lytic replication [48]. Finally, we tested whether the S13 mutation affected EBV protein expression (Fig 6F). Western blot assay revealed that the EBNA1 expression was not distinguishably different among four types of HEK293-EBV cells regardless of induction. The BZLF1 expression in induced Mt BART(-)·S13⁻ HEK293-EBV cells was slightly higher than that of Wt BART(-)·S13⁺ HEK293-EBV cells. In similar perspective, This BZLF1 expression was not directly associated to the increases of extracellular EBV genome copy numbers in induced Mt BART(-)·S13⁻ HEK293-EBV cells. Taken together, these data indicate that the S13 mutation plays a regulatory role in establishing EBV latency within the episome, regulating BART miRNA expression, and inducing EBV abortive lytic reactivation.

## Investigation of epigenetic regulation mediated by EBV S13 locus

We further investigated the epigenetic effects of the S13 mutation in EBV genomes using four types of BART(+/-)·S13 HEK293-EBV cells and performing ChIP-qPCR assay for CTCF, RNAP II, H3K4me3 histone, and H3K9me3 histone (Fig 7). Primer sets for the ChIP-qPCR assay were selected from CTCF binding sites identified by a previous ChIP-Seq assay (S2 Table). Compared with the Mt BART(+/-)·S13⁻ HEK293-EBV cells, we found interesting results for Mt BART(+/-)·S13⁻ HEK293-EBV cells. Firstly, enrichment of CTCF at S15 and S16 loci reduced up to 95% and 67%, respectively, in Mt BART(-)·S13⁻ HEK293-EBV cells, but increased up to 44% and 52%, respectively, in Mt BART(+)·S13⁻ HEK293-EBV cells (Fig 7A). Secondly, enrichment of RNAP II at the S13 locus decreased up to 47% in Mt BART(-)·S13⁻ HEK293-EBV cells but increased up to 64% in Mt BART(+)·S13⁻ HEK293-EBV cells (Fig 7B). Thirdly, enrichment of H3K4me3 histone at the S13 locus reduced up to 49% in Mt BART(-)· S13⁻ HEK293-EBV cells but increased up to 43% in Mt BART(+)·S13⁻ HEK293-EBV cells (Fig 7C). Finally, enrichment of H3K9me3 histone at the S13 locus increased up to 65% in Mt

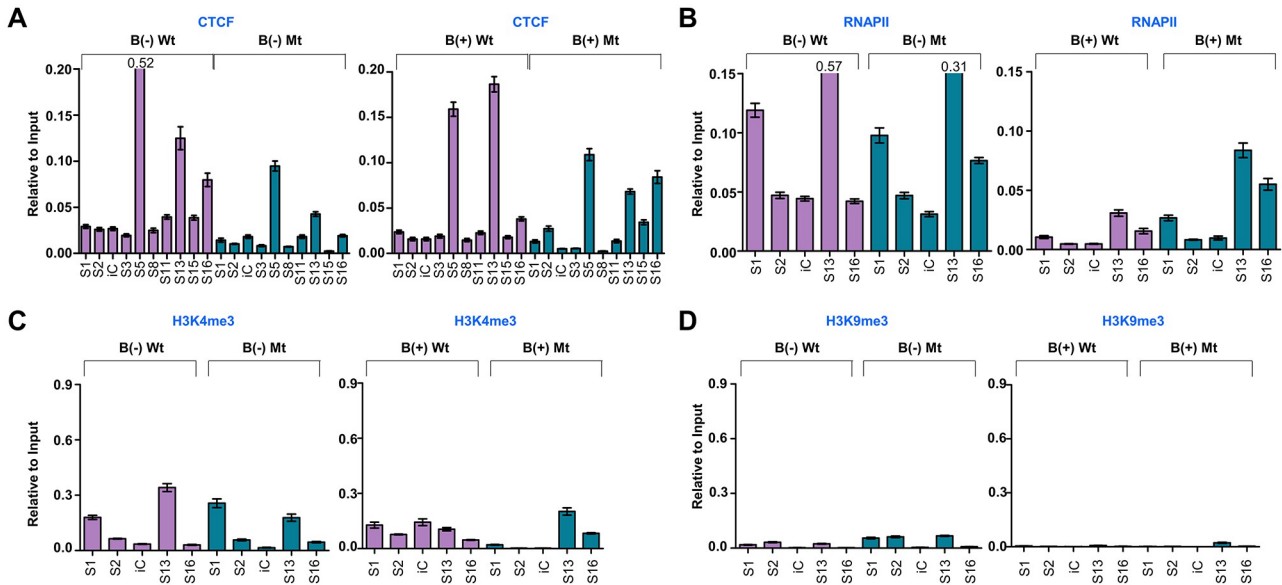

**Fig 7. Analysis of epigenetic factors nearby CTCF binding sites in HEK293-EBV BART S13 cells.** ChIP-qPCR assay was conducted to identify epigenetic factors nearby CTCF binding sites on EBV genome. **A)** CTCF ChIP-qPCR assay. Enrichment of CTCF at S1-S16 sites identified by ChIP-Seq assay was compared between BART(+/-)·S13$^+$ HEK293-EBV cells and Mt BART(+/-)·S13$^-$ HEK293-EBV cells. iC stands for an internal control, selected from the region (EBV 38173 ~ 38272) where CTCF did not bind to the EBV genome of SNU719 cells. **B)** RNAP II ChIP-qPCR assay. Enrichment of RNAP II at S1-S16 sites identified by ChIP-Seq assay was compared between BART(+/-)·S13$^+$ and Mt BART(+/-)·S13$^-$ HEK293-EBV cells. **C)** H3K4me3 histone ChIP-qPCR assay. Enrichment of H3K4me3 histone at S1-S16 sites identified by ChIP-Seq assay was compared between BART(+/-)·S13$^+$ and Mt BART(+/-)·S13$^-$ HEK293-EBV cells. **D)** H3K9me3 histone ChIP-qPCR assay. Enrichment of H3K9me3 histone at S1-S16 sites identified by ChIP-Seq assay was compared between BART(+/-)·S13$^+$ and Mt BART(+/-)·S13$^-$ HEK293-EBV cells. In panels, experiments were independently repeated two times, and data are represented as mean ± SD.

BART(-)·S13$^-$ HEK293-EBV cells but did not change in Mt BART(+)·S13$^-$ HEK293-EBV cells (Fig 7D). Collectively, these data indicate that epigenetic regulation favorably supports the biological role of the S13 locus as a transcriptional regulator of EBV BART miRNA expression.

## Investigation of EBV infectivity regulated by S13 locus

Four types of EBV bacmid genome (Wt BART(+)·S13$^+$, Mt BART(+)·S13$^-$, Wt BART(-)·S13$^+$, and Mt BART(-)·S13$^-$) were tested for EBV infectivity in HEK293 cells. To this end, each type of BART(+/-)·S13$^{+/-}$ HEK293-EBV cell was transfected with pcDNA3-BZLF1 and pcDNA3--BALF4. Transfected BART(+/-)·S13$^{+/-}$ EBVs were harvested three days post-transfection. Harvested viruses were tested for infectivity in HEK293 cells (S6 Fig). Interestingly, Mt BART (+/-)·S13$^-$ EBVs were severely defective in their infectivity to HEK293 cells compared to Wt BART(+/-)·S13$^+$ EBVs (Fig 8A). For statistical analysis, we selected at least three different regions on the culture surfaces of each type of BART(+/-)·S13$^{+/-}$ HEK293-EBV cells and then counted all GFP spots in the selected regions. The GFP spots in Mt BART(+/-)·S13$^-$ HEK293-EBV cells were compared with those in Wt BART(+/-)·S13$^+$ HEK293-EBV cells (Fig 8B). Mt BART(+/-)·S13$^-$ EBVs were significantly defective in infecting HEK293 cells compared to Wt BART(+/-)·S13$^+$ EBVs. Since HEK293-EBV cells produced differently extracellular EBV genome copies depending on cell types (Fig 6A), it was necessary to analyze EBV infectivity on normalized condition with equal numbers of EBV virions. Thus, we normalized total GFP spot per each type of HEK293-EBV cells with 1 x 10$^6$ EBV virions, compared total GFP spots among BART(+/-)·S13$^{+/-}$ EBVs, and analyzed infectivity of each EBV to HEK293 cells. As shown (Fig 8C), EBVs containing the S13 mutation appeared less infectious than Wt EBVs

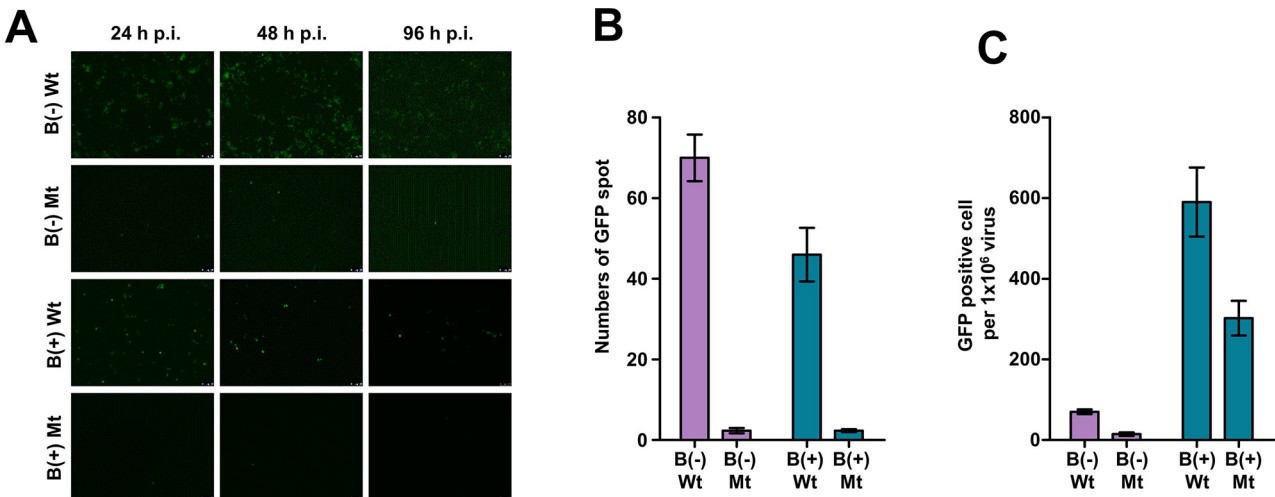

**Fig 8. EBV infection assay. A)** Infection assays of BART(+/-)·S13⁺ EBV and Mt BART(+/-)·S13⁻ EBV using HEK293 cells. EBV Infection was observed at different time-periods such as 0, 24, 48, and 96 h post-infection. **B)** Relative quantification of GFP spots on three regions of surface of cultured HEK293 cells indicating where EBVs infected. B(-)Wt, B(-)Mt, B(+)Wt, and B(+)Mt stand for BART(-)·S13⁺ EBV, Mt BART(-)·S13⁻ EBV, BART(+)·S13⁺ EBV, and Mt BART(+)·S13⁻ EBV, respectively. **C)** Normalization of the quantified GFP spots produced from HEK293-EBV cells by 1 x 10⁶ EBV virion. In panels, experiments were independently repeated two times, and data are represented as mean ± SD.

regardless of the 11.8-kb *Eco*RI C fragment (BART). Taken together, these results indicate that the S13 locus, and not the BART transcript, is required to maintain complete EBV infectivity.

## Investigation of changes in S13 CTCF-mediated chromatin interactions

To examine whether the S13 locus is involved in the formation of DNA loops, a 4C-Seq assay was conducted using four types of BART(+/-)·S13$^{+/-}$ HEK293-EBV cells in a same manner described above. We examined the genomic association with the S13 locus in BART(+/-)· S13$^{+/-}$ HEK293-EBV cells (Fig 9A). In similar, filtrated reads were adjusted to 3,000 reads for the analysis of each cell (S5 Table) and same DNA-DNA association should appear at least twice in the 4C-seq analysis. Comparing with Wt BART(-)·S13$^{+}$ genome, the S13 locus in Mt BART(-)·S13$^{-}$ genome was uniquely associated with 2~7-kb region (near S1), 71~76-kb region (near S9), 100~105-kb region (near S12), 122~132-kb region, 155~160-kb region, and 162~167-kb region (near S13) (Fig 9A). In addition, the S13 locus in Mt BART(+)·S13$^{-}$ genome was uniquely linked to 2~7-kb region (near S1) and 81~86-kb region (near S11) whose genomic linkage was not found in Wt BART(+)·S13$^{-}$ genome (Fig 9A). These results indicated that the S13 locus plays a key role in EBV genomic association along with the *Eco*RI C fragment (BART). Beside the S13 locus, we further investigated EBV genomic associations of other loci in similar manner except adjusting filtered reads and limiting DNA-DNA associations as described above. Total filtrated reads from all tested viewpoints (BART-1, BART-2, LMP, Qp, OriP, FR) were adjusted to 7,000,000 for the analysis of each cell (S4 Table). All DNA-DNA associations (including at least one association) were determined in parallel to compare among EBV genomes in HEK293-EBV cells (S4 Table). First, we could not find any genomic association with the BART locus in both Wt BART(-)·S13$^{+}$ and Mt BART(-)·S13$^{-}$ EBV genomes in the 4C-Seq assay since the BART-1 viewpoint does not exist in both genomes (Fig 9B). However, BART loci in Wt BART(+)·S13$^{+}$ and Mt BART(+)·S13$^{-}$ EBV genomes were associated similarly with the 8~13-kb region (near S2), 45~50-kb region (near S5), 100~105-kb region (near S11/S12), 120~125-kb region, and 155~160-kb region (near S16) (Fig 9B).

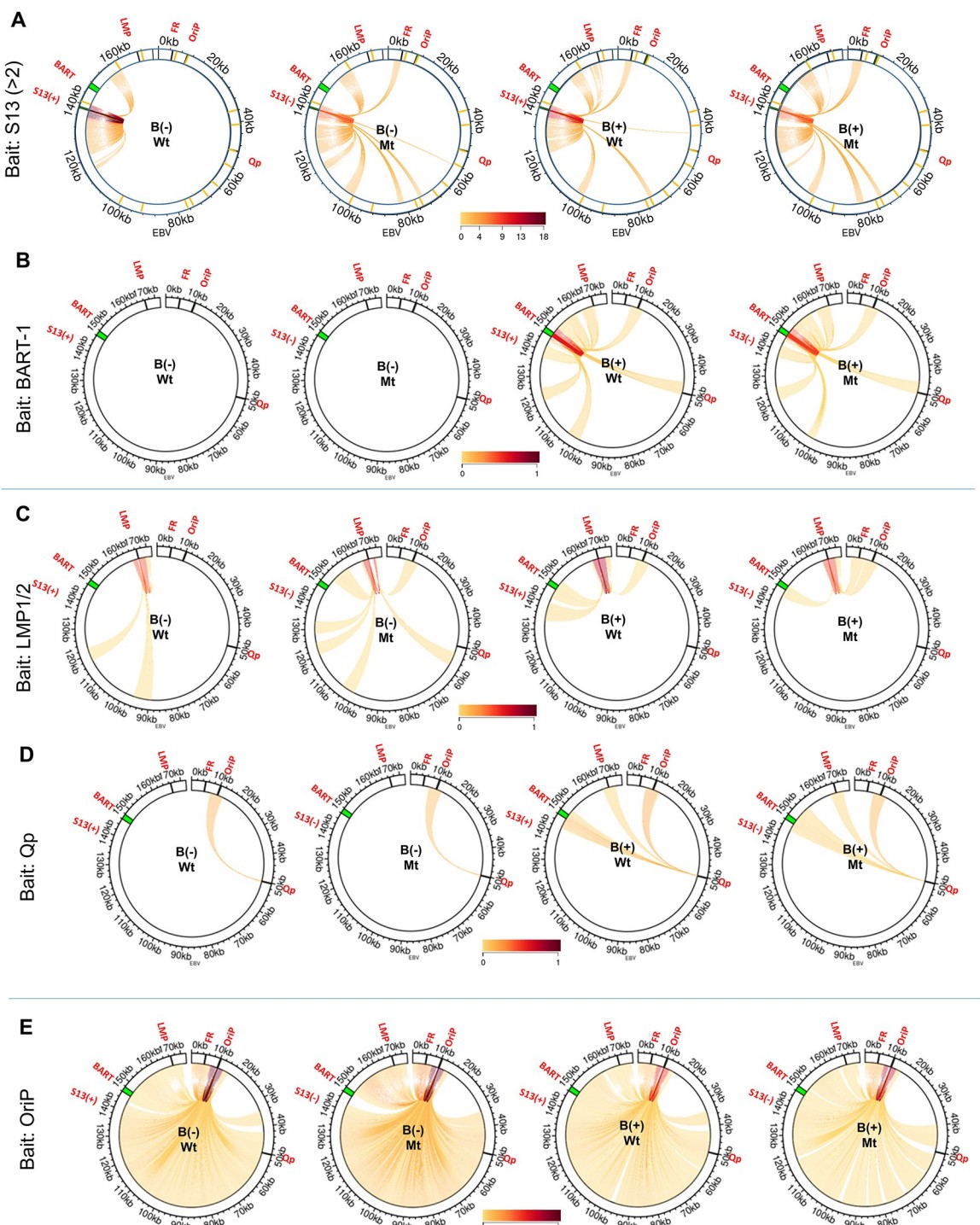

**Fig 9. Confirmation of chromatin interaction affected by BARTp CTCF BS mutation.** Significant genomic associations of target viewpoint regions in HEK293-EBV genomes identified by 4C-Seq assays. **A)** To compare EBV genomic associations with the S13 locus, we adjusted the 4C-seq read numbers from S13 viewpoint primer set to 3,000 in the 4C-seq analysis of Wt BART(+/-)·S13+ and Mt BART(+/-)·S13- cells (S5 Table). **B-E)** To compare EBV genomic associations with other loci such as BART-1, BART-2, LMP1/2, Qp, OriP, and FR, total read numbers from all target viewpoints were adjusted to 7,000,000 in each cell (S4 Table). 4C-seq assays revealed all interactions of BART-1 **(B)**, LMP1/2 **(C)**, Qp **(D)**, and OriP **(E)** regions in Wt BART(+/-)·S13+ and Mt BART(+/-)·S13- EBV genomes. Locations of viewpoint primer sets were posted in S3 Table; S13 (EBV 138504 ~ 138987), BART-1 (EBV 147357 ~ 147490), LMP1/2 (EBV 166480 ~ 166661), Qp (EBV 50278 ~ 51579), OriP (EBV 9918 ~ 10302) and FR (EBV 4731 ~ 4961).

Overall, we could not find any significant difference in BART associations between Wt BART (+)·S13⁺ and Mt BART(+)·S13⁻ EBV genomes (Fig 9B). Secondly, the LMP1/2 locus in Mt BART(-)·S13⁻ EBV genome was associated uniquely with the 8~13-kb region (near S2), 52~57-kb region (near S5/S6), 95~100-kb region (near S11), 120~125-kb region, and 128~133-kb region (Fig 9C). These LMP1/2 associations were not found in Wt BART(-)·S13⁺ EBV genome. Furthermore, the LMP1/2 locus made selective interactions with 135~135-kb region (near S13) in Wt BART(-)·S13⁺ EBV genome and with 2~7-kb region (near S1) in Mt BART(-)·S13⁻ EBV genome (Fig 9C). Thirdly, the Qp loci in both Mt BART(-)·S13⁻ and Wt BART(-)·S13⁺ EBV genomes were similarly associated with only the 8~13-kb region (near S2) (Fig 9D). In similar, the Qp loci in both Mt BART(+)·S13⁻ and Wt BART(+)·S13⁺ EBV genomes were also similarly linked to the 8~13-kb region (near S2), 145~150-kb region (near S14), and 155~160-kb region (near S16) (Fig 9D). Overall, we could not find any significant difference in Qp associations between Wt BART(+/-)·S13⁺ and Mt BART(+/-)·S13⁻ EBV genomes (Fig 9D). Fourthly, the OriP loci in both Mt BART(+/-)·S13⁻ and Wt BART(+/-)·S13⁺ EBV genomes were innumerably associated with multiple loci in the EBV genomes (Fig 9E). Last, the FR loci in both Mt BART(+/-)·S13⁻ and Wt BART(+/-)·S13⁺ EBV genomes were also enormously linked to multiple loci in the EBV genomes (S8 Fig). We speculated that these similar association patterns were due to the close proximity of locations of two viewpoints.

To verify EBV genomic associations with the S13 locus defined in 4C-seq assay, 3C-PCR assays with BART(+/-)·S13 HEK293-EBV cells were conducted to consolidate chromatin interactions and confirm DNA associations defined by previous 4C-seq assays. Multiple chromatin interactions between the S13 locus and other EBV genomic loci were observed in all four types of BART(+/-)·S13 HEK293-EBV cells (S9A Fig). The viewpoint primer sets for the 3C-PCR assay were selected from the CTCF binding sites identified by the previous ChIP-Seq assay (S4 Table). In Wt BART(+/-)·S13⁺ HEK293-EBV cells, a 135-kb region (near S13) was associated with a 3-kb region (near S1), 35-kb region (near S3), and 167-kb region (near S16) (S9A Fig). In Mt BART(+/-)·S13⁻ HEK293-EBV cells, the 135-kb region was associated with only the 35-kb region (S9A Fig). In general, the S13 mutation caused slight defects in the formation of DNA-DNA associations, regardless of the presence of 11.8-kb EcoRI C fragment. A 3-kb region (near S1) also associated with the 167-kb region (near S16) and 88-kb region (near S11) in BART(+/-)·S13⁺ HEK293-EBV cells (S9B Fig). However, the association of the 3-kb region with the 167-kb region was very weak in Mt BART(-)·S13⁻ HEK293-EBV cells, and was not observed in Mt BART(+)·S13⁻ HEK293-EBV cells (S9B Fig). As a control, we conducted PCR assays with unligated XhoI-digested DNA samples from Wt BART(+/-)·S13⁺ HEK293-EBV cells (S10A Fig) and Mt BART(+/-)·S13⁺ HEK293-EBV cells (S10B Fig) under the same conditions as those used for PCR with corresponding ligated XhoI-digested BART (+/-)·S13 HEK293-EBV DNA samples; no false positive 200~500 bp band was amplified from the 3C-PCR primer sets.

The S13 mutation might cause more damage to the DNA interaction (between 3-kb and 167-kb regions) in the presence of the 11.8-kb EcoRI C fragment than in the absence of the EcoRI C fragment (Fig 10). Some DNA associations in 3C-PCR assays were not well consistent to corresponding results in 4C-seq assay. However, taken together, these all results were enough to suggest that a functional S13 locus is essential for the formation of a cluster of EBV genomic loci that play a regulatory role in EBV BART miRNA expression and abortive lytic reactivation.

## Discussion

Gene expression regulation requires the integration and coordination of various signals across the genome. EBV gene expression is controlled at multiple levels, including transcription

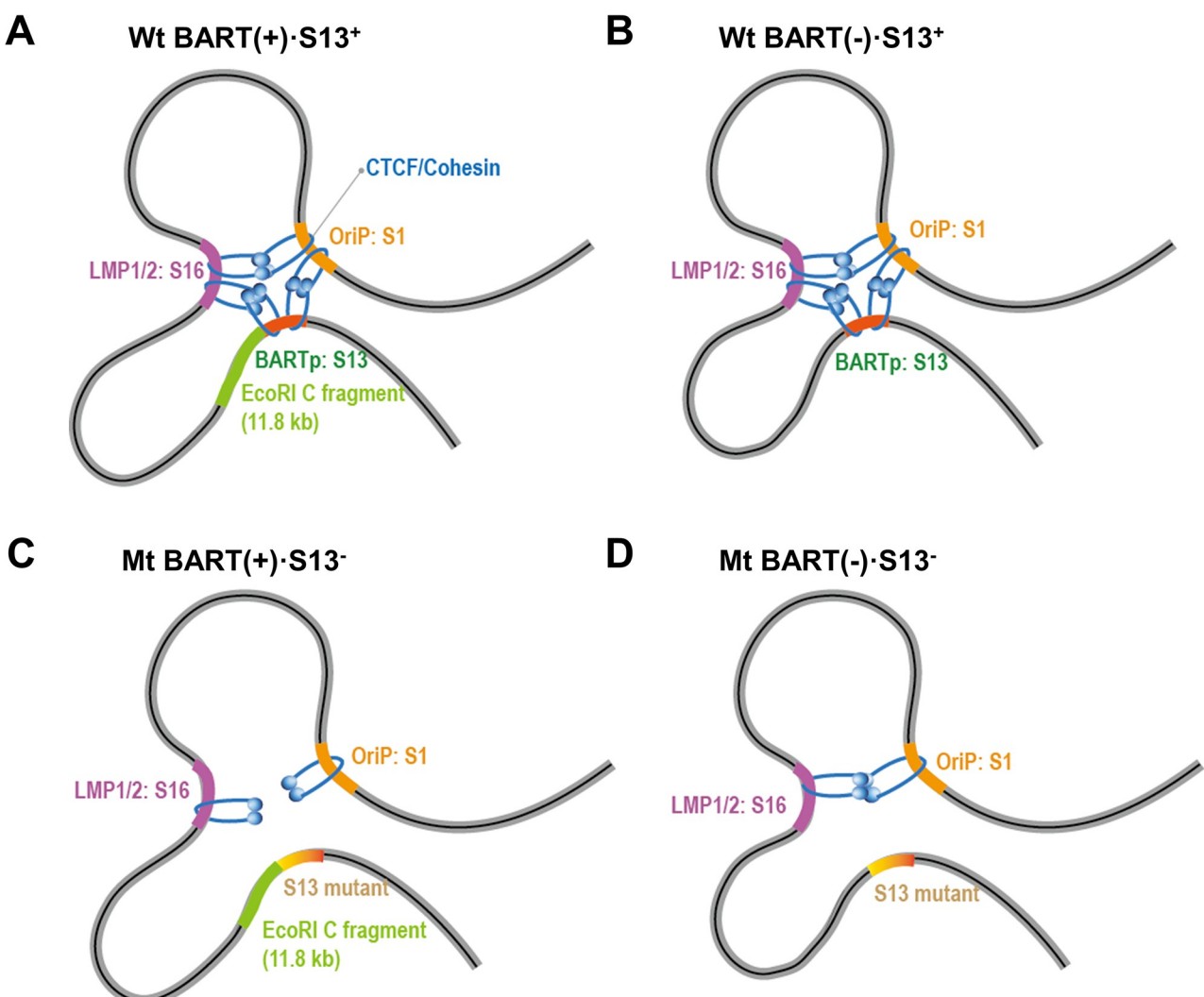

**Fig 10. Simple diagrams explaining the different effects of the S13 mutation on the formation of EBV chromatin loops.** Deletion of the S13 locus causes distorted CTCF distribution in EBV genomes. Compared to BART(+)·S13+ EBV genome **(A)** and BART(-)·S13+ EBV genome **(B)**, CTCF was enriched at the S15 and S16 loci in the Mt BART(+)·S13- EBV genome **(C)**, but less CTCF was enriched at the S15 and S16 loci in the Mt BART(-)·S13- EBV genome **(D)**. Consequently, deletion of the S13 locus severely damaged S1 and S16 interactions in the Mt BART(+)·S13- EBV genome than in the Mt BART(-)·S13- EBV genome. BARTp represents a CTCF-binding site on the BART promoter.

factor binding, initiation and elongation, RNA processing, and epigenetic modifications that collectively control gene expression during latent and lytic phases [6]. CTCF-mediated chromatin interactions play a key role in regulating EBV gene expression. CTCF binding sites in OriP (S2 [49]), Qp (S5 [43]), and LMP1/2 (S16 [42]) have been characterized for their functional roles in regulating EBV gene expression. Most of these studies focused on EBV genomes in BL or lymphoblastoid cell lines (LCLs). Although the 11.8-kb *Eco*RI C fragment is well-conserved in EBV-associated Raji, Mutu, and GC1 cells, few genetic studies have been conducted to define miRNAs in the 11.8-kb *Eco*RI C fragment [47]. In particular, the functional role of a CTCF binding site (S13 locus) in the EBV BART promoter has not been characterized.

Here, we first showed a cluster of DNA loops formed by the interaction of S13 locus with several CTCF binding sites in the EBV genome. Firstly, the S13 locus was associated with at least 3-kb, and 167-kb regions in the B95-8 and SNU719 EBV genomes. Secondly, a functional

S13 locus was required to form a stable DNA loop between the 3-kb (OriP locus) region and 167-kb (LMP1/2 locus) regions. Thirdly, the S13 locus was found to be an important site for transcriptional regulation of EBV BART miRNAs. Fourthly, we found that the functional S13 locus was necessary for the full capacity of EBV infection. Last, we observed that the functional S13 locus was involved in maintaining EBV latency and suppressing EBV abortive lytic reactivation. All these findings indicate that the S13 locus coordinates the EBV life cycle by forming chromatin interactions with other CTCF binding sites in the viral genome.

CTCF has been implicated in cohesin-chromatin interactions by forming DNA loops [50]. Long-distance DNA interactions are essential for mediating communication between promoter and enhancer elements. EBV OriP is a transcriptional enhancer of Cp and LMP1 promoters, whose action mechanism involves the CTCF-mediated DNA loop structure [51]. In the present study, we examined the biological roles of CTCF bound to the EBV BART promoter in B95-8 cells, SNU719 cells, and HEK293-EBV cells.

We found that deletion of the CTCF binding site (S13 locus) from the EBV BART promoter resulted in decreased enrichment of CTCF on S15 and S16 (LMP1/2) loci in Mt BART(-)·S13⁻ EBV genome. However, the deletion caused greater enrichment of CTCF on the S15 and S16 loci in Mt BART(+)·S13⁻ EBV genome. Furthermore, the chromatin interaction of OriP (S1 locus) and LMP1/2 (S16 locus) turned more severely defective in Mt BART(-)·S13⁻ EBV genome than Mt BART(+)·S13⁻ EBV genome. The distorted distribution of CTCF may significantly alter chromatin interactions between the S1 and S16 loci (Fig 10). Therefore, both the functional S13 locus and the 11.8-kb BART region might play important roles in forming CTCF-mediated DNA loops between the OriP and LMP1/2 loci.

Another biological role of the CTCF-bound S13 locus may be to link the LMP1/2 locus and the OriP locus into a cluster of DNA loops. This genomic clustering partly accounts for the role of CTCF in maintaining latent EBV infection. Similar to this study, we previously observed that chromatin interactions between the LANA and RTA loci could form a chromatin complex via CTCF during latent KSHV infection [52]. In the KSHV life cycle, these CTCF-mediated interactions are disrupted during KSHV lytic reactivation [52]. Furthermore, KSHV spontaneously initiated its lytic life cycle when the chromatin interaction between LANA and RTA loci was artificially dismantled. In our study, deletion of CTCF bound to the S13 locus upregulated the expression of EBV lytic genes and overproduced extracellular EBV genomes in Mt BART(+)S13⁻ HEK293-EBV cells, indicating that CTCF bound to the S13 locus facilitates EBV latency. CTCF has been implicated as a chromatin insulator and a boundary factor [53, 54]. CTCF can prevent epigenetic drift by blocking heterochromatin formation in the EBV Qp region [43]. Deleting the CTCF binding site in EBV LMP1/2 resulted in the disruption of the DNA association between OriP and LMP1/2 loci, an increase in H3K9me3 histone and DNA methylation at the LMP1 promoter region, and severe reduction of EBV latent infection [42]. In the present study, the deletion of the CTCF binding site in the EBV BART promoter (S13 locus) downregulated the expression of BART miRNAs, decreased CTCF enrichment at S15 and S16 loci, reduced RNA polymerase bound to S13 and S16 loci, and enhanced H3K9me3 histone bound to S13 in the Mt BART(-)·S13⁻ EBV genome. As a result, Mt BART(-)·S13⁻ EBVs lost their ability to establish latent infection. Therefore, the functional S13 locus is involved in stabilizing CTCF-mediated DNA loops between the OriP and LMP1/2 loci, and consequently maintains EBV latency.

Mutation of the S13 locus in Mt BART(-)·S13⁻ EBV genome increased the *BZLF1* expression (Fig 6D–6F) but greatly reduced extracellular EBV genome copy numbers (Fig 6A), suggesting that the S13 mutation induces EBV abortive lytic reactivation [48]. It was known that the abortive lytic infection is present in the pre-latent phase and converges to latent infection. Main features of abortive lytic infection are not productive for infectious virions and not

required the BZLF1 protein [48]. Thus, the phenotype of Mt BART(-)·S13⁻ EBV genome was alike to that of EBV genomes in the abortive lytic infection. Moreover, the S13 locus plays an essential factor in transcribing miR-BART1 and miR-BART15 when an 11.8-kb *Eco*RI C fragment is present. In contrast, the S13 locus is required to suppress the upregulation of miR--BART1 and miR-BART15 when an 11.8-kb *Eco*RI C fragment is absent. Concomitantly, the S13 locus works a dual transcriptional role depending on surrounding cis-acting factors. Hence, we propose that the S13 locus works along with the 11.8-kb *Eco*RI C fragment as a regulator to maintain EBV latency and suppress uncoordinated EBV abortive lytic reactivation, resulting in failure to produce infectious EBV virions.

This phenotype of Mt BART(+)·S13⁻ EBVs was similar to that of the LMP1/2 CTCF Mt EBVs (with deletion of CTCF binding site on LMP1/2 locus). In addtion, the S13 locus simultaneously interacted with the S16 (CTCF binding site on LMP1/2) and S1 (CTCF binding site on OriP) loci in the EBV genome. Based on these results, one possible mechanism related to the S13 mutation is that CTCF bound to the BART promoter might work together with CTCFs bound to LMP1/2 and OriP to block the spread of heterochromatin possibly induced by GC-rich respective DNA of the EBV terminal repeat (TR) region. We suggest that CTCF bound to the S13 locus is one of three essential factors for forming a DNA loop cluster, along with CTCFs bound to the S16 and S1 loci. However, the molecular mechanism downstream of the CTCF and BART promoter interaction that coordinate the association between S16 locus (LMP1/2) and S1 locus (OriP) in blocking the spread of heterochromatin for EBV gene expression remain to be understood.

Recent studies using 3D genomic methods examined EBV interactions with host chromosomes using various cell types [55, 56]. Another related study used capture Hi-C to analyze KSHV looping DNA interactions during latency and lytic phase reactivation and found that DNA loops were organized around highly active RNAP II promoters, especially the viral PAN non-coding RNAs [57]. The present study used 4C-Seq analysis to define DNA looping interactions within EBV genomes in both marmoset B-cell and GC cells, focusing on the regions controlling BART transcript expression. Our study shows that like the KSHV PAN promoter, EBV BART promoter is an organizing hub for the EBV genome.

Along with the binding data of RNAP II and H3 histones on the EBV S13 locus, CTCF-mediated chromatin interactions are likely to account for the most DNA loops formed within EBV genomes. It is also possible that the EBV association with the host chromosome contributes to some aspects of the EBV chromosome conformation. Although our high-resolution 4C analysis revealed extensive conformational structures of EBV genomes during EBV latency in marmoset B-cells and human GC, further studies are required to resolve some of these more complicated functions of CTCF and to better understand how EBV has exploited CTCF binding sites to coordinate gene regulation and genome propagation during latent infection.

## Materials and methods

### Cell lines and reagents

HEK293 cells and four types of HEK293-EBV BART(+/-)·S13 bacmid harboring cell lines were cultured in Dulbecco's Modified Eagle Medium (DMEM; Hyclone, Pittsburgh, PA, USA) supplemented with 10% Fetal Bovine Serum (FBS; Hyclone, Marlborough, MA, USA), antibiotics/antimycotics (Gibco, Waltham, MA, USA), and Glutamax (Gibco, Waltham, MA, USA) at 37°C under conditions of 5% $CO_2$ and 95% humidity in a $CO_2$ incubator. HEK293 cells were transfected with four types of BART(+/-)·S13 EBV plasmids and then selected using hygromycin B (400 μg/mL) (Wako, Japan). The GC cell line SNU719 (EBVaGC) was purchased from the Korean Cell Line Bank (Seoul, Korea), and EBV-producing marmoset B-cell

line B95-8 cells were kindly gifted by Dr. Lieberman (Wistar Institute, PA, USA). Both cell lines were cultured in RPMI 1640 medium (Hyclone, Pittsburgh, PA, USA) supplemented with 10% FBS, antibiotics/antimycotics, and Glutamax at 37˚C under conditions of 5% $CO_2$ and 95% humidity.

## Microscale thermophoresis (MST) assay

A wild-type (Wt) S13 49-mer primer set targeting EBV 138946–138994 region was designed, and the forward primer was labeled with 5-carboxyfluorescein (5-FAM) (Fig 2B). We designed a parallel mutant (Mt) S13 49-mer set as a counter partner to the Wt S13 49-mer primer set. Mt S13 49-mer contained several point mutations in the region spanning EBV 138,963–138,987, which were expected to disrupt CTCF binding to the Wt S13 49-mer primer set (Fig 2B). The forward Wt S13 49-mer primer (10 µL) was mixed with 10 µL of 100 µM reverse Wt S13 49-mer primer. The mixture was paired by placing it in a thermocycler, which was programmed to start at 95˚C for 2 mins and then gradually cool to 25˚C for 45 mins. In parallel, 10 µL of 100 µM forward Mt S13 49-mer primer was mixed and paired with 10 µL of 100 µM reverse Wt S13 49-mer primer. The resultant paired Wt and Mt S13 49-mer primer sets were used as the target DNA in the MST assay. His-tagged CTCF proteins were purified from Sf9 cells transfected with the CTCF expression plasmid using the baculovirus expression system. Either purified CTCF proteins or two types of Sf9 lysates were used as ligand proteins in the MST assay. To block non-specific binding to CTCF proteins or Sf9 lysates, 2000 ng sonicated salmon sperm DNA was incubated in 20 µL reaction solution with the CTCF protein or Sf9 lysates for 60 min on ice. Then, we made CTCF proteins or Sf9 lysates bind to Wt and Mt S13 49-mer primer sets for 60 min on lice using following reaction mixture: 5 µL 4X EMSA buffer (400 mM KCl, 80 mM HEPES, 0.8 mM EDTA, 80% glycerol, pH 8.0) freshly added 1 mM DTT, target DNA (125 nM), 2.5 µL ligand protein (CTCF proteins, Sf9 lysates: 0 µM, 1.56 µM, 3.12 µM, 6.25 µM, 12.5 µM), and 8.5 µL sterile water. Finally, Nanotemper Monolith NT.115 (Munich, Germany) was used to analyze the binding of CTCF and its target oligo, as recommended by the manufacturer.

## Real-time quantitative PCR (qPCR) assay

Precipitated DNA was quantified using a real-time quantitative PCR (qPCR) assay with SYBER Green in FastStart Essential DNA Green Master (Roche, Basel, Switzerland). Each resultant DNA was diluted using nuclease-free water and analyzed in triplicate for the expression of EBV-associated genes and CTCF binding sites. The PCR mixture (20 µL contained 5 µL template DNA, 0.5 µM of each primer, and 10 µL of Master SYBER Green 1 mix (Roche, Basel, Switzerland). Features of the primer sets used in ChIP-qPCR are listed in S2 Table. The following thermal cycling conditions were used: 95˚C for 10 min; 45 cycles of 95˚C for 10 s, 55˚C for 10 s, and 72˚C for 10 s; 95˚C for 5 s; followed by 65˚C for 1 min.

## PCR assay

PCR assays were performed as previously described [58] in a 25 µL reaction volume containing 5 µL of 5× reaction mix, 5 µL of 5× TuneUp solution, 1 µL of Taq-plus polymerase, and 2.5 µL of 10 pmol forward and reverse primers. The following thermal cycling conditions were used: 95˚C for 3 min, 30 cycles of 95˚C for 10 s, Tm (specific to primer sets) for 30 s, and 72˚C for 30 s, followed by 72˚C for 10 min. Features of the primer sets used in the PCR assay are listed in the S5 Table. The PCR assays were performed on a TaKaRa PCR Thermal Cycler (TaKaRa, Kyoto, Japan) and the amplicons were resolved on a 1.2% agarose/TBE gel.

## Construction of recombinant EBV bacmid

Site-directed mutations were introduced at the CTCF binding site on the BART promoter (S13 locus) in BART(+) or BART(-) EBV bacmids to generate Mt BART(+/-)·S13⁻ EBV bacmids. Mt BART(+/−)·S13⁻ EBV bacmids were generated using a two-step red-mediated recombination method. Primers used for generating PCR products of the kanamycin resistance gene (Kanʳ) from the pEPKanS3 plasmid were as follows: forward primer 5'-GCA TCT TTC TAA CCA GTA GGG GCC TCC ACC TAG GTG CTT TGT TAA TCT TTA GTG TAT ATA TAT ATA TAT ATA TGG GTA CCC CTA TCC TAC AAC CAA TTA ACC AAT TCT GAT TAG-3' and the reverse primer 5'- ACA GGG ATT ATC AAG ACA AGG AGC TCC GGT AGG ACC TAT AGG ATA GGG GTA CCC ATA TAT ATA TAT ATA TAT TAT ACA CTA AAG ATT AAC AAA GGA TGA CGA CGA TAA GTA GGG ATA- 3'. The resultant PCR products were composed of the Kanʳ gene with the I-SceI site flanked by 60 bp downstream and upstream of the EBV S13 sequence surrounding the CTCF mutation sequence. These PCR products were electroporated into GS1783 competent cells containing BART(+/−)·S13⁺ EBV bacmids for first round of homologous recombination. Mt BART(+/-)·S13⁻ EBV bacmid harboring Kanʳ gene was recovered by positive selection, characterized by restriction enzyme digestion, and transformed into GS1783 I-SceI-inducible competent cells. The Kanʳ gene was removed from EBV bacmid-Kanʳ during second round homologous recombination and negative selection was performed to generate complete Mt BART(+/-)·S13⁻ EBV bacmids. Finally, the CTCF mutation was confirmed via restriction enzyme digestion and sequencing of the homologous recombination site in Mt BART(+/−)·S13⁻ EBV bacmids.

## Transfection

Four types of BART(+/-)·S13 (BART(+)S13⁺, Mt BART(+)S13⁻, BART(-)S13⁺, and Mt BART (-)S13⁻) EBV bacmids were transfected into HEK293 cells using a Neon transfection system (Invitrogen, Carlsbad, CA, USA). Cells ($5 \times 10^4$) were resuspended in 100 μL serum-free media containing 5 μg of each Mt recombinant bacmid. Electroporation was conducted using a Neon electroporator (Invitrogen) set at 1350 V for 30 ms and one pulse. Cells were placed in a medium supplemented with 10% FBS for 48 h post-transfection. Next, transfected cells were selected using 40 μg/mL hygromycin B for two weeks. Finally, four types of HEK293-EBV cells were established and named BART(-)·S13+, Mt BART(-)·S13-, BART(+)·S13+, and Mt BART (+)·S13- HEK293-EBV cells.

## Western blotting assay

Western blot analysis was performed using four types of BART(+/-)·S13 (BART(+)·S13⁺, Mt BART(+)·S13⁻, BART(-)·S13⁺, and Mt BART(-)·S13⁻) HEK293-EBV cells. Before conducting Western blot assay, we induced EBV lytic reactivation by treating these cells with TPA (20 ng/mL) and NaB (3 mM) for 48 h. HEK293-EBV cells ($5 \times 10^6$) were lysed using 100 μL RIPA lysis buffer (Promega, WI) supplemented with 1 μL proteinase inhibitor and 10 μL phenylmethylsulfonylfluoride. The cell lysates were fractionated using a Bioruptor sonicator (5 min, 30 sec on/off pulses). Cell lysates were loaded onto 8% sodium dodecyl sulfate polyacrylamide electrophoresis gels and subjected to western blot analysis using the following primary antibodies against EBV proteins (1:1000 dilution): anti-EBV EBNA1 (Santa Cruz Biotechnology (SCB), Santa Cruz, CA, USA), anti-EBV BZLF1 (SCB), and anti-beta Actin (SCB). The secondary antibodies used were horseradish peroxidase (HRP)-conjugated sheep anti-mouse IgG (Genetex, Irvine, CA, USA), HRP-conjugated donkey anti-rabbit IgG (Genetex), and HRP-conjugated goat anti-rat IgG (Bethyl Laboratories, Montgomery, TX, USA).

## CTCF ChIP-Seq analysis

A ChIP-Seq assay against CTCF was performed with $5 \times 10^7$ B95-8 and SNU719 cells per sample and crosslinked using formaldehyde. Crosslinked B95-8 and SNU719 cell lysates were sonicated to obtain DNA fragments of ~100–500 bp length. Immunoprecipitation was performed with 10 μg of either rabbit anti-CTCF (Millipore, Burlington, MA, USA) or control rabbit IgG (GeneTex, Irvine, CA, USA) and incubated overnight with antibody-coated Dynabeads protein A/G (Invitrogen). Incubated beads were washed five times with ChIP-Seq wash buffer (50 mM HEPES, pH 7.5, 500 mM LiCl, 1 mM EDTA, 1% NP-40, 0.7% Na-Deoxycholate, 1x protease inhibitors) for five times, then washed once with 50 mM NaCl in TE buffer. Immunoprecipitated DNA was eluted using ChIP-Seq elution buffer (50 mM Tris-HCl, pH 8, 10 mM EDTA, 1% SDS), reverse-crosslinked at 65˚C, treated with RNase A (0.2 mg/mL) and proteinase K (0.2 mg/mL), purified with phenol and chloroform, and subjected to qPCR validation. Validated ChIP samples were isolated by agarose gel purification, ligated to primers, and subjected to Illumina-based sequencing using the manufacturer's protocol (Illumina, San Diego, CA, USA). ChIP-Seq reads were mapped to the EBV wild-type reference genome (NC 007605) using the Bowtie software. The mac2 tool was used for peak calling [59], and MEME was used to draw diverse LoGo of CTCF binding sites identified by our CTCF ChIP-Seq assay [60].

## ChIP-qPCR assay

ChIP assays for CTCF, RNAP II, H3K4me3, and H3K9me3 were performed using $3 \times 10^6$ B95-8 cells, SNU719 cells, and BART(+/-)·S13 (BART(+)·S13$^+$, Mt BART(+)·S13$^-$, BART(-)·S13$^+$, Mt BART(-)·S13$^-$) HEK293-EBV cells per sample, according to the crosslinking chromatin immunoprecipitation (X-ChIP) protocol provided by Abcam (Cambridge, UK) with a slight modification. Before crosslinking, we induced EBV lytic reactivation by treating these cells with TPA (20 ng/mL) and NaB (3 mM) for 48 h. According to the manufacturer's protocol, a Bioruptor (BMS, Korea) was used to sonicate the genomic DNA. Sonicated cell lysates were subjected to immunoprecipitation with antibodies against CTCF (07–729, Millipore), RNAPII (ab26721, Abcam), H3K4me3 (07–442, Millipore), H3K9me3 (07–472, Millipore), and normal rabbit IgG (GTX35035, Genetex). The precipitates were incubated with ChIP elution buffer (1% SDS, 100 mM NaHCO$_3$); next, samples were reverse-crosslinked at 65˚C overnight and purified using Promega columns. Purified DNA was analyzed using RT-qPCR. ChIP DNA values were calculated as a proportional expression over isotype-specific input DNA values of each primer set.

## 3C-PCR and 4C-Seq assays

Chromatin confirmation capture (3C)-PCR assays were performed as previously described [56]. Briefly, $1 \times 107$ B95-8, SNU719, and four types of BART(+/-)·S13 HEK293-EBV cells were fixed with 1% paraformaldehyde for 10 min at 37˚C. Nuclei were permeabilized by incubation with 0.5% SDS at 62˚C for 10 min, digested with 100 units of XhoI (New England Biolabs (NEB), Ipswich, MA, USA), and ligated into the nucleus, followed by an in situ 3C protocol [43]. After reversal of crosslinks, 3C DNA products were prepared and subjected to PCR assay using 0.5 μg, 5 μg, and 50 μg of 3C DNA products as template to determine DNA associations within EBV genomic loci.

As previously reported, the circular chromatin confirmation capture (4C)-Seq assay was performed [56]. Briefly, $1 \times 107$ B95-8, SNU719, and four types of BART(+/-)·S13 HEK293-EBV cells were fixed with 1% paraformaldehyde for 10 min at 37˚C. Nuclei were permeabilized by incubation with 0.5% SDS at 62˚C for 10 min, digested with 100 units of MboI (NEB), and ligated into the nucleus, following the in situ Hi-C protocol [61]. The ligated DNA product was further digested with 100 units of Csp6I (NEB) and re-ligated. For each viewpoint

primer, 10–100 ng DNA was amplified by PCR. Features of the viewpoint primer sets used in the 4C-Seq assay are shown in S3 Table. All samples were sequenced using a NovaSeq 6000 100 bp paired read. The 4C-Seq assay was performed using all viewpoints in biological replicates for B95-8, SNU719, and four types of BART(+/-)·S13 HEK293-EBV cells.

## 4C-Seq data analysis

The 100-bp sequence paired-end reads were trimmed and grouped using cutadapt (version 2.10) based on the sequence from the 4C bait location. Reads were aligned to the EBV genome (NC_007605) using Bowtie2 (version 2.2.3) and an iterative alignment strategy. Reads with low mapping quality (MapQ < 10) and those mapped to human repeat sequences were removed. The total aligned reads for each $i$-th position of the non-overlapping 10 kb window ($N_i$) were calculated. The $P$-values were converted using the Poisson formula:

$$Pi = 1 - \sum j = 0Ni\lambda e - \lambda/j!Pi = 1 - \sum j = 0Ni \ \lambda e - \lambda/j! \tag{1}$$

Where $\lambda$ is equal to the average of reads for each 10 kb window (except EBV-aligned reads). Significant peaks were defined using the subcommand bdgpeakcall of MACS2 software (version 2.1.1) with the following parameters: at least $P$-value $< 10^{-5}$ (option -c 5), minimum length of 20 kb (option -l 20000), and maximum gap of 10 kb (option -g 10000). The total significant peak number for each 10 kb of bait positions (BART-1, BART-2, OriP, Qp, LMP1/2) was counted (S4 Table). To analyze DNA associations per each type of subject cells, we adjusted 4C-seq read numbers from read targets (viewpoints) up to 7,000,000 (S4 Table) and compared in parallel their DNA associations among subject cells. For S13 locus, we adjusted 4C-seq read numbers to 3,000 in additional analysis (S5 Table).

## Southern blot assay

Genomic DNA (gDNA) was extracted from four types ($5 \times 10^6$ BART(+/-)·S13 (BART(+)·S13$^+$, Mt BART(+)·S13$^-$, BART(-)·S13$^+$, Mt BART(-)·S13$^-$) of HEK293-EBV cells, as recommended by NEB. Next, 40 μg extracted gDNA was digested with *Bam*HI overnight. The digested gDNA was purified using phenol-chloroform-isoamyl alcohol (25:24:1) and 20 μg purified gDNA was separated on 0.8% agarose gel and transferred to a nylon membrane (Roche). After UV crosslinking, nylon membrane was prehybridized at 42˚C for 2 h and hybridized at 42˚C overnight in DIG Easy Hyb buffer (Roche) with digoxigenin (DIG)-labeled probe. The labeled probe was generated by PCR with EBV-specific sequence primers, whose target was a part of the EBV *BLLF1* gene (EBV 78803–79522). PCR to generate EBV-BLLF1 probe was performed using PCR DIG Probe Synthesis Kit (Roche) with the following primers and cycle conditions: F: 5-CCA GGC CCA AAA GGC AGT CA-3, R: 5-TCA CCA CCG GAG AGG AGC AA-3; and 95˚C for 2 min; 10 cycles of 95˚C for 10 s, 60˚C for 30 s, 72˚C for 2 min; 20 cycles of 95˚C for 10 s, 60˚C for 30 s, 72˚C for 2 min 20 s, and finally 72˚C for 7 min. The resultant amplicon size was 719 bp, as confirmed by electrophoresis. After hybridization with this probe, the nylon membrane was washed and blocked using the DIG wash and block buffer kit (Roche). The nylon membrane was further incubated with anti-DIG-AP (Roche) in a blocking solution at 4˚C overnight and stained with CDP-Star (Roche) for 5 min in the dark. The signals were detected using a ChemiDoc imaging system (Bio-Rad).

## EBV infection study

Four types of BART(+/-)·S13 (BART(+)·S13$^+$, Mt BART(+)·S13$^-$, BART(-)·S13$^+$, and Mt BART(-)·S13$^-$) HEK293-EBV cells ($6.3 \times 10^5$) were separately seeded in 6-cm culture dishes.

The next day, HEK293-EBV B(+/-)·S13 cells were transfected with pCDNA3-BZLF1 and pCDNA3-BALF4 using TurboFect Transfection Reagent (Thermo Scientific, Waltham, MA, USA). The medium was replaced with fresh medium one day post-transfection. The medium was harvested three days post-transfection, clarified by performing two rounds of centrifugation at 3000 rpm and passed through once using 0.45 μm filter (Sartorius Stedim Biotech, Gottingen, Germany) and kept for EBV infection. Next, HEK293 cells were plated onto a 6-well culture plate. The next day, the spent culture medium was removed from the HEK293 cells. A 40 μm cell strainer (FALCON, Corning, NY, USA) was placed over onto HEK293 cells. Then we infected with EBV the HEK293 cells by adding the clarified old medium to the HEK293 cells and incubating for 24 h. After 24 h incubation, we replaced the clarified old medium with fresh medium and observed GFP signals on the infected HEK293 cells.

### Analysis of BART miRNA expression

Total RNA was isolated from B95-8 cells, SNU719 cells, and four types of BART(+/-)·S13 HEK293-EBV cells using TRIzol reagent (Invitrogen) according to the manufacturer's instructions. Before extracting RNA, we induced EBV lytic reactivation by treating these cells with TPA (20 ng/mL) and NaB (3 mM) for 48 h. Extracted total RNA was reverse-transcribed using the miRCURY LNA RT Kit (Qiagen) per manufacturer's instructions. For real-time PCR, the miRCURY LNA SYBR Green PCR Kit (Qiagen) was used following manufacturer's instructions on a LightCycler 96 (Roche). The primers used were as follows: #YP00205809 (Qiagen) for ebv-miR-BART2-5p; #YP00205853 (Qiagen) for ebv-miR-BART11-5p; #YP00205734 (Qiagen) for ebv-miR-BART1-5p;, #YP00205800 (Qiagen) for ebv-miR-BART15;, #YP00205763 (Qiagen) for ebv-miR-BART6-5p; and #YP00203907 (Qiagen) for U6 snRNA. The amount of BART miRNA expressed was normalized against U6 snRNA used as an endogenous control and analyzed using the $2^{-\Delta\Delta Ct}$ method. Each reaction was performed in triplicate.

### Statistical analysis

Statistical analysis was performed using both Kruskal-Wallis test (nonparametric test) as prior test and Dunn's test as post test to compare among phenotypes of three more cells. In addition, paired t test was also used to compare between phenotypes of two cells. Results with P-values (one-tailed) <0.05 (95% confidence) were considered statistically significant.

### Supporting information

**S1 Fig. Brief 4C experimental procedure and 4C-plots for FR. A)** Four genomic loci (BART, OriP, Qp, and LMP1/2) were used as baits. The viewpoint primer sets are listed in S3 Table. **B)** 4C-seq assays revealed all interactions of FR regions with other loci in EBV genomes in B95-8 cells and SNU719 cells. The FR viewpoint primer sets were located at EBV 4731–4751 and EBV 4941–4961.
(TIF)

**S2 Fig. Analysis of EBV genomic associations by 3C-PCR assay using B95-8 and SNU719 cells.** Significant genomic associations of target viewpoint regions in B95-8 and SNU719 EBV genomes were identified by 3C-PCR assays. To verify EBV genomic associations defined by 4C-seq analysis, 3C-PCR assay was conducted using B95-8 **(A)** and SNU719 **(B)** cells. Linked associations between the bait and target regions were amplified as PCR products in the 3C-PCR assay. 3C DNA products were prepared and subjected to PCR assay using 0.5 μg (label-1), 5 μg (label-2), and 50 μg (label-3) of 3C DNA products as template to determine DNA associations within EBV genomic loci. The bait region was the 135-kb locus adjacent to

S13. Target region sizes were 3-kb (near S1), 35-kb (near S3), 49-kb (near S5), 88-kb (near S11), and 167-kb (near S16), respectively. The tiny arrow indicates the PCR product suggesting association between the 135-kb region and one of the target regions in the 3C-PCR assay. 3-kb & 167-kb association was tested by 3C-PCR assay with OHK649 and OHK648 primer set, 135-kb & 3-kb association with OHK728 and OHK649 primer set, 135-kb & 35-kb association with OHK728 and OHK687 primer set, 135-kb & 49-kb association with OHK728 and OHK683 primer set, 135-kb & 65-kb association with OHK728 and OHK689 primer set, 135-kb & 88-kb association with OHK728 and OHK691 primer set, and 135-kb & 167-kb association with OHK728 and OHK648 primer set.
(TIF)

**S3 Fig. Negative control experiments in 3C-PCR assay for EBV genomic associations in B95-8 and SNU719 cells.** Negative experiments in 3C-PCR assay for analysis of EBV genomic associations were conducted using unligated *Xho*I-digested B95-8 (A) and SNU719 (B) DNA samples under the same conditions as the PCR assay with ligated *Xho*I-digested B95-8 and SNU719 DNA samples. Unligated *Xho*I-digested B95-8 and SNU719 DNA samples were not subjected to T4 DNA ligase mediated ligations. Unligated DNA samples were subjected to PCR assay using 0.5 μg (label-1), 5 μg (label-2), and 50 μg (label-3) of unligated DNA samples as template to determine false-positive amplification from primer sets in 3C-PCR assays. PCR primer sets used in analyzing ligated DNA samples were equally used exploited to negative experiments using unligated DNA samples. This PCR assay with unligated DNA samples were considered as negative control experiment to assess false positivity of 3C-PCR assay with ligated DNA samples.
(TIF)

**S4 Fig. Sequencing of linked DNA with S13 locus.** DNA fragments amplified from 3C-PCR assay were verified to link with S13 locus (EBV genome 135-kb region). **A)** 397-bp DNA (S13 locus) fragment indicating the linkage of 135-kb region to 3-kb region (S1 locus) was cloned to pGEM-T vector. The cloned T vector was subjected to sequence insert DNA fragments using T7 or Sp6 primer sets. **B)** 364-bp DNA fragment indicating the linkage of 135-kb region to 167-kb region (S16 locus) was cloned to pGEM-T vector. The cloned T vector was subjected to sequence insert DNA fragments using T7 or Sp6 primer sets.
(TIF)

**S5 Fig. Generation of Mt BART(+/−)·S13⁻ HEK293-EBV cells. A)** Schematic diagram of BART(+/-)·S13⁺ EBV bacmids. A CTCF-binding site on the BART miRNA promoter was identified as BARTp CTCF BS. **B)** Sequences introduced at the points of recombination to introduce site-directed mutations at the S13 locus (EBV 138963–138987). **C)** Confirmation of the site-directed mutation in the S13 locus in the Mt BART(+)·S13⁻ EBV bacmid by Sanger sequencing. The Mt BART(-)·S13⁻ EBV bacmid was also confirmed by Sanger sequencing. **D)** Gel electrophoresis to check stabilities of EBV genome in EBV bacmids following *EcoRI* digestion; BART(-)·S13⁺ EBV bacmid (lane 1, -w), Mt BART(-)·S13⁻ EBV bacmid (lane 2, -m), BART(+)·S13⁺ EBV bacmid (lane 3, +w), and Mt BART(+)·S13⁻ EBV bacmid (lane 4, +m). Bands generated by *EcoRI* digestion of both BART(-)·S13⁺ and Mt BART(-)·S13⁻ EBV bacmids are marked on lanes 1 and 2 of the gel by the top two arrows. Other bands specific to *EcoRI* digestion of both BART(+)·S13⁺ and Mt BART(+)·S13⁻ EBV bacmids are marked on lanes 3 and 4 of the gel by the bottom two arrows. **E)** Establishment of BART(+/-)·S13 (HEK293-EBV BART(-)·S13⁺, HEK293-EBV BART(-)·S13⁻, HEK293-EBV BART(+)·S13⁺, and HEK293-EBV BART(+)·S13⁻) HEK293-EBV cells. HEK293 cells were transfected with all four types of BART (+/-)·S13 (BART(-)·S13⁺, BART(-)·S13⁻, BART(+)·S13⁺, and BART(+)·S13⁻) EBV plasmids

and selected using hygromycin B to establish all four types of BART(+/-)·S13 HEK293-EBV cells. GFP expression was determined 40 d after hygromycin B selection for several passages. Established BART(+/-)·S13 HEK293-EBV cells maintained GFP expression even after several passages. **F)** Southern blot analysis was conducted to test if BART(+/-)·S13 EBV bacmids established their episomes in HEK293 cells. Briefly, genomic DNAs from all four types of BART(+/-)·S13 HEK293-EBV cells were digested with *Eco*RI, purified with Phenol/chloroform/isoamyl alcohol solution treatment, and run on a gel. DNA probe was formed from EBV 78803 to 79522, and genomic DNA from HEK293 cells was used as a negative control. **G)** Confirmation of the deletion of the CTCF binding site (S13 locus) on the BART miRNA promoter. A CTCF ChIP-qPCR assay was conducted to confirm the abruption of the functional S13 locus (CTCF binding) in Mt BART(-)·S13⁻ cells and Mt BART(+)·S13⁻ HEK293-EBV cells. **H)** Comparison of *RPMS1* expression patterns between BART(+/-)·S13⁺ and Mt BART(+/-)·S13⁻ HEK293-EBV cells. RT-qPCR assay was conducted to measure amounts of *RPMS1* transcripts in HEK293-EBV cells. In panels, experiments were independently repeated two times, and data are represented as mean ± SD.
(TIF)

**S6 Fig. Schematic diagram of EBV infection assay.** BART(+/-)·S13⁺ and Mt BART(+/-)·S13⁻ HEK293-EBV cells were transfected with pCDNA3-BZLF1 and pcDNA3-BALF4. At three days post-transfection, supernatants of all four types of transfected BART(+/-)·S13 HEK293-EBV cells were harvested and added to HEK293 cells freshly cultured on a 6-well plate where a cell strainer was placed to remove cell debris. EBV in harvested supernatants were allowed to infect HEK293 cells for 24 h. After infection, EBV-encoded GFP in infected HEK293 cells was detected over a series of time points.
(TIF)

**S7 Fig. Absolute quantification of intracellular and extracellular EBV genome copy numbers.** Absolute quantification of relative intracellular and extracellular EBV genome copy numbers. qPCR assay with *EBNA1* primer set (EBV 96778–96979, EBV 96827–96845) was conducted to measure EBV genome copy numbers in all four types of HEK293-EBV cells which were either uninduced or induced with TPA and NaB. B(-)Wt, B(-)Mt, B(+)Wt, and B(+)Mt stand for HEK293-EBV cells of BART(-)·S13⁺, BART(-)·S13⁻, BART(+)·S13⁺, BART(+)·S13⁻, respectively. Both absolute intracellular and extracellular EBV genome copy numbers were evaluated based *EBNA1* Ct values. Relation equation between absolute DNA amounts and *EBNA1* Ct values was first defined and then applied to calculate DNA concentration (μg/μL) per length of template (bp) using a website called Copy Number Calculator of Technology NetWorks (https://www.technologynetworks.com/tn/tools/copynumbercalculator). In panels, experiments were independently repeated two times, and data are represented as mean ± SD. Statistical analysis was performed using both Kruskal-Wallis test (nonparametric test) as prior test and Dunn's test as post test to compare among phenotypes of subject cells.
(TIF)

**S8 Fig. 4C-plots for FR in BART(+/-)·S13 EBV HEK293-EBV cells.** 4C-seq assays revealed all interactions of FR regions with other loci in EBV genomes in BART(+/-)·S13 HEK293-EBV cells. The FR viewpoint primer sets were located at EBV 4731–4751 and EBV 4941–4961.
(TIF)

**S9 Fig. Analysis of EBV genomic associations by 3C-PCR assay using BART(+/-)·S13 HEK293-EBV cells.** To verify EBV genomic associations defined by 4C-seq analysis, 3C-PCR assay was conducted using the 135-kb locus **(F)** and 3-kb locus **(G)** in Wt BART(+/-)·S13⁺ and

Mt BART(+/-)·S13⁻ HEK293-EBV cells. Linked associations between the bait and target regions were amplified as PCR products in the 3C-PCR assay. 3C DNA products were prepared and subjected to PCR assay using 0.5 μg (label-1), 5 μg (label-2), and 50 μg (label-3) of 3C DNA products as template to determine DNA associations within EBV genomic loci. The bait region was the 135-kb locus adjacent to S13. Target region sizes were 3-kb (near S1), 35-kb (near S3), 49-kb (near S5), 88-kb (near S11), and 167-kb (near S16), respectively. The tiny arrow indicates the PCR product suggesting association between the 135-kb region and one of the target regions in the 3C-PCR assay. 3-kb & 167-kb association was tested by 3C-PCR assay with OHK649 and OHK648 primer set, 135-kb & 3-kb association with OHK728 and OHK649 primer set, 135-kb & 35-kb association with OHK728 and OHK687 primer set, 135-kb & 49-kb association with OHK728 and OHK683 primer set, 135-kb & 65-kb association with OHK728 and OHK689 primer set, 135-kb & 88-kb association with OHK728 and OHK691 primer set, and 135-kb & 167-kb association with OHK728 and OHK648 primer set.
(TIF)

**S10 Fig. Negative control experiments in 3C-PCR assay for EBV genomic associations in BART(+/-)·S13 HEK293-EBV cells.** Negative experiments in 3C-PCR assay for analysis of EBV genomic associations were conducted using unligated *Xho*I-digested BART(+/-)·S13 HEK293-EBV DNA samples under the same conditions as the PCR assay with ligated *Xho*I-digested BART(+/-)·S13 HEK293-EBV DNA samples. Negative control experiments in 3C-PCR assay were conducted using the 135-kb locus **(A)** and 3-kb locus **(B)** in Wt BART (+/-)·S13⁺ and Mt BART(+/-)·S13⁻ HEK293-EBV cells. Unligated *Xho*I-digested BART(+/-)· S13 HEK293-EBV DNA samples were not subjected to T4 DNA ligase mediated ligations. Unligated DNA samples were subjected to PCR assay using 0.5 μg (label-1), 5 μg (label-2), and 50 μg (label-3) of unligated DNA samples as template to determine false-positive amplification from primer sets in 3C-PCR assays. PCR primer sets used in analyzing ligated DNA samples were equally used exploited to negative experiments using unligated DNA samples. This PCR assay with unligated DNA samples were considered as negative control experiment to assess false positivity of 3C-PCR assay with ligated DNA samples.
(TIF)

**S1 Table. Sequences identified by ChIP-Seq assay as CTCF binding site in EBV genomes.**
(DOCX)

**S2 Table. Locations and sequences of primer sets used in ChIP-qPCR assay.**
(DOCX)

**S3 Table. Locations and sequences of viewpoint primer sets used in 4C-Seq assay.**
(DOCX)

**S4 Table. Adjustment of filtrated 4C-sequencing reads.**
(DOCX)

**S5 Table. Adjustment of filtrated 4C-sequencing reads for S13 locus.**
(DOCX)

**S6 Table. Locations and sequences of primer sets used in 3C-PCR assay.**
(DOCX)

**S7 Table. Quantification of band intensity by densitometry\* of Southern blot analysis.**
(DOCX)

## Author Contributions

**Conceptualization:** Inuk Jung, Hyojeung Kang.

**Data curation:** Hideki Tanizawa.

**Funding acquisition:** Sun Hee Lee, Kyoung-Dong Kim, Hyosun Cho, Hyojeung Kang.

**Investigation:** Sun Hee Lee, Kyoung-Dong Kim, Miyeon Cho, Sora Huh, Seong Ho An, Donghyun Seo, Kyuhyun Kang, Minhee Lee, Hideki Tanizawa.

**Supervision:** Hyosun Cho.

**Writing – original draft:** Inuk Jung, Hyosun Cho, Hyojeung Kang.

**Writing – review & editing:** Inuk Jung, Hyosun Cho, Hyojeung Kang.

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
