## [Decision Letter · Decision Letter 0]

3 Jun 2022

Dear Dr. Kang,

Thank you very much for submitting your manuscript "Characterization of a new CCCTC-binding factor binding site as a dual regulator of Epstein-Barr virus latent infection" for consideration at PLOS Pathogens. As with all papers reviewed by the journal, your manuscript was reviewed by members of the editorial board and by several independent reviewers. In light of the reviews (below this email), we would like to invite the resubmission of a significantly-revised version that takes into account the reviewers' comments.

Thank you for sharing your thorough experiments studying the role of CTCF binding in regulating EBV chromatin and transcription. The careful reviewers have noted many questions and suggestions regarding the arguments presented. The most important point of improvement for the revision is an increased focus on rigor and reproducibility. Please clarify concerns about statistics, technical vs. biological replicates, data presentation, and use of the appropriate controls. If any critical elements are missing, they must be included in the revision. These critiques particularly, though not exclusively, apply to the Western blot, 4C-seq, ChIP-qPCR, and 3C-PCR.

We cannot make any decision about publication until we have seen the revised manuscript and your response to the reviewers' comments. Your revised manuscript is also likely to be sent to reviewers for further evaluation.

Sincerely,

JJ Miranda

Guest Editor

PLOS Pathogens

Erik Flemington

Section Editor

PLOS Pathogens

Kasturi Haldar

Editor-in-Chief

PLOS Pathogens

orcid.org/0000-0001-5065-158X

Michael Malim

Editor-in-Chief

PLOS Pathogens

orcid.org/0000-0002-7699-2064

Thank you for sharing your thorough experiments studying the role of CTCF binding in regulating EBV chromatin and transcription. The careful reviewers have noted many questions and suggestions regarding the arguments presented. The most important point of improvement for the revision is an increased focus on rigor and reproducibility. Please clarify concerns about statistics, technical vs. biological replicates, data presentation, and use of the appropriate controls. If any critical elements are missing, they must be included in the revision. These critiques particularly, though not exclusively, apply to the Western blot, 4C-seq, ChIP-qPCR, and 3C-PCR.

Reviewer's Responses to Questions

**Part I - Summary**

Reviewer #1: This manuscript describes the interaction of chromatin insulator CTCF with EBV episomes and identifies a novel CTCF binding site in the BART miRNA promoter. The authors go on to characterize the function of this CTCF binding site by 1. comparing EBV isolates that with and without a deletion in the region of the novel CTCF binding site and 2. by site directed mutagenesis of EBV bacmids to ablate CTCF binding. The findings are of interest and timely and the study has the potential to add new knowledge to that already reported. However, there are several technical issues (highlighted below) that need to be resolved. In addition, the manuscript is poorly written in parts and therefore difficult to follow. In my opinion, the manuscript cannot be accepted for publication in it's current state.

Reviewer #2: In this manuscript the authors identified CTCF binding site on the EBV genome in two different EBV+ cell lines that carry different EBV genomes. While SNU719 EBV is intact, there is a 11.8kb deletion near to S13 CTCF binding site in B95-8 EBV. The authors used 4C-seq, 3C-PCRs and EBV mutagenesis to explore the biological role of CTCF binding in the BART miRNA promoter region (S13 locus) regarding EBV gene regulation. The results suggest that the CTCF binding in S13 locus reduces RNA polymerase II binding at S13 and also reduces EBV gene expression. In contrast, mutation of CTCF-binding site in S13 reduces EBV infectivity suggesting that CTCF in S13 locus may be required for EBV replication, or virus production or infection. While the manuscript includes a large amount of work and data, it requires some additional experiments in some cases to support their conclusion or to clarify ambiguous statements.

Reviewer #3: CTCF insulators are important transcriptional regulators in eukaryotic cells and it is becoming increasingly clear that these insulators-together with cohesin- play important roles in controlling transcription of DNA viruses through the formation of 3D chromatin loops, including in EBV. In this current work, Lee et al expand on the current knowledge of 3D loops in EBV by identifying additional long-range interactions between sequences mapping near the BART region of EBV and the S13 locus. These data suggest that the S13 locus is involved in BART miRNA expression. The data presented by Lee et al are extensive. For example, they perform ChIP seq experiments to show that 16 sites in both B95-8 and SNU719 genomes are enriched in CTCF, they show biological function through functional assays (conventional ChIP, gene expression, mutational analyses, etc) and showed that mutations in S13 resulted in decreased integration into cellular genomes, and a reduction in EBV infectivity. There are a few areas of the manuscript that are more descriptive than others. For example, it would be nice to describe mechanistically why CTCF, RNA Pol II, and histones are more enriched at S13 in B95-8 and not SNU719. What effect does this have on the viral phenotype? Viral fitness? replication? reactivation? Finally, while the data is compelling, I do not think that the authors truly show that the long-range interaction coordinates BART miRNA expression and infectivity- more mechanistic evaluations in cell types other than HEK cells should be done to definitively prove this. My one major concern with the manuscript is the absolute lack of detail in the number of replicates done for each experiment and the lack of statistics that identify what is statistically different. Specific comments are below:

**Part II – Major Issues: Key Experiments Required for Acceptance**

Reviewer #1: Deletion in EBV genome in B95-8 should be annotated in figure 1A. Annotation of key EBV genes and enhancer/promoter features in this figure would also be useful. Coverage in the deleted region in B95-8 in the IgG ChIP-Seq track in SNU719 cells appears to be markedly reduced in the same region. How do the authors explain this when there is no deletion in this region in these cells?

Fig2A – it is impossible to tell the quality of the purified CTCF protein from the western blot provided for the following reasons: it is tightly cropped so that spurious bands cannot be seen (in fact there are multiple bands in the WB section that is shown), there is no molecular weight marker so the size of the protein is not indicated and, most importantly, the authors do not provide a Coomassie stained gel image to show a single, purified band.

In fig 2B the authors show the mutations introduced into the S13 CTCF binding site. These are extensive mutations. Is this a coding region, or are there any other known features in this region of the EBV genome. How can the authors be sure they are not affected another viral function with such dramatic changes? They apparently show no change in RPMS1 expression in Fig S3B but this is just a sequence alignment, not expression data. What is the biological effect of the sequence differences between WT and mutant? This becomes most important in the interpretation of the infectivity assays shown in Figure 8.

The labelling of Fig 2C is confusing. The top two panels are data collected following addition of Sf9 lysate (?) but the fluorescence traces still labelled with CTCF. The authors should calculate Kd values from these traces (ideally from multiple experiments +/- S.D.) rather than relying on the visual interpretation of the shift in the trace upon addition of protein.

Figure 5 needs to be explained more fully. Are Bio1 and Bio2 replicates? What do the numbers next to the legend for each Circos plot mean? Annotation of the location of the CTCF binding sites identified (S1-16) would be helpful for interpretation. The 3C experiments shown in Fig5D and Figure 9 would benefit from a figure explaining the method and location of primers used. What are the lanes labelled 1, 2 and 3. There do not appear to be any positive or negative controls for this experiment. Were the products Sanger sequenced? Why are there multiple bands visible for each experiment?

Figure 6B – molecular weight markers should be labelled on the Southern blot. To draw the conclusion stated in lines 286-288, the authors need to quantify band intensity by densitometry of several repetitions of the Southern blot analysis.

It is unclear what the error bars indicate in the ChIP-qPCR experiments shown in figures 3 and 7. Is error calculated from multiple repetitions of the qPCR of a single ChIP experiment, or from the mean of multiple independent ChIP experiments. I suspect they are the former as there are little variation in the data. If so, I am not confident that the differences observed between cell types and EBV bacmid genomes are meaningful. These experiments should be repeated multiple times and reported as individual experiments in the supplementary or mean and error of independent experiments.

What does figure 8A show? This panel does not add anything meaningful.

Lines 384-387. How do the authors conclude that the interactions in EBV episomes are ‘simultaneous’. Presumably there are multiple copies of the viral episome in a single cell. They are analysing the episome conformations in the entire population. There may be a mixture of different forms that cannot be unpicked from the bulk population.

Reviewer #2: Fig 1 CTCF ChIP-seq was done in two different cell lines but there is no mention in the manuscript about their comparison or how much this data is aligned with previous CTCF ChIP-seq on EBV genome. What is Y axis on the graph?

Why is the sequence of S4 and S14 identical?

Line 132 Multiple Em…What is Em?

How similar is the CTCF binding site logo to the consensus CTCF sites identified in previous studies?

Fig 2A MW is missing from the immunoblot. Is this lysate or purified CTCF on the immunoblot? In Fig 2C, the bound oligo/CTCF curve should not be on the top while unbound should be the lower curve? What is [-] on the Y axis? What does the blue and pink vertical lines represent? Their labels are missing. It would be better to use different line colors for the different CTCF conc.

Fig 3. The authors wanted to investigate the binding of epigenetic factors and RNAPII on the BART miRNA region (S13) but instead the graphs show the enrichment of different transcription-related factors at different CTCF binding site including S13. Can they test for more genomic sites around S13 spanning the BART miRNA promoter in more detail? Also, I recommend that they add a BART miRNA promoter region schematic and indicate where the ChIP-qPCR primers are located.

Line 170: delete the word histone after H3K4me3 and H3K9me3. They are histone marks.

I agree with the observation that CTCF, RNAPII and H3K4me3 are more enriched at S13 and S16 in B95-8 than in SNU719. While the 11.8kb EcoRI fragment may play a role in this discrepancy but another possibility, which was not excluded, that quality of chromatin and thereby that of ChIPs are different. In fact, in Fig 1A CTCF ChIP-seq seems to have higher background or was less efficient in SNU719 compared to B95-8.

Line 180-182: I do not see how the ChIP data in Fig 3 indicate that S13 locus plays a role in the expression of BART miRNAs by coordinating the enrichment of RNAPII, CTCF, and histones. Why histones? No histones were measured here. In addition, there was no functional analysis, and EBV is defective near to S13 in B95-8. Their conclusion would be valid if the 11.8kb fragment were added back to the EBV genome in B95-8 and they would see different BART miRNA expression and changes in RNAPII and in the enrichment of histone mark.

Fig 4. Is the viral gene expression difference between B95-8 and SNU719 restricted to BART miRNAs or there is difference in the expression of other EBV genes as well?

Fig 5 is supposed to show chromatin interaction of S13 locus with other parts of the EBV genome. Instead, the 4C-seq shows the interactions of 8-9kb upstream BART regions with the EBV genome. Although the data is interesting but it is unclear why they did not use S13 locus as a viewpoint for the 4C-seq?

In Fig 5D some of the 3C-PCRs do not seem specific but generate several bands. How do we know which band corresponds to what specific DNA sequence?

Line 262. The bacmid DNAs were transfected and not transduced into HEK293.

Fig S3E. If these images were taken using confluent cell cultures, most of the cells are not GFP+ even after 40 days of hygromycin B selection. If this the case, it would question the meaning of every experiment that were performed with these cell lines.

Fig 6A. Were the cells induced to measure extracellular EBV DNA? If not, how can EBV DNA be detected in supernatant? Or B(+) EBV-transfected cells are lytic?

Fig 6B. How is it determined based EBV BLLF1 gene southern blot which EBV genome is integrated into the human genome and which not?

Line 296. “The S13 locus is required for normal…” sentence needs to be revised using the name of the right EBV clones.

Fig 6D. Were these cells lytically induced or spontaneous/abortive lytic reactivation was measured?

How the differential expression of genes in S13 mutant virus are connected to EBV 3D structure and looping between S13 locus and other parts of the EBV genome?

Line 304-306. Conclusion of experiments in Fig 6 needs to be revised. The mutation of S13 site was used to determine what the role of S13 in wild type EBV. Thus, the results show the role of S13 locus in the regulation of BART miRNA expression and inhibition of lytic cycle and not the function of S13 mutation.

Line 313-314. “Compared with the mutant…” sentence needs to be revised using the name of the correct EBV clones. The same EBV clones are compared to the same virus clones?

Line 324-325. The conclusion sentence needs to be revised to make it more specific and meaningful. Fig 6 data, which showed increased lytic gene expression with S13 mutation in wt EBV, is in line with increased enrichment of RNAPII at S13 in B(+) Mt in Fig 7. Therefore, Fig 6 and 7 together indicate that CTCF-binding in the S13 locus favors less RNAPII binding, which correlates with less viral gene transcription. This is not the case when we compare B(-)wt with B(-)mut indicating the 11.8kb EcoRI fragment may affect the function of S13 locus in viral gene regulation.

General comment: it is unclear how many times the ChIPs were done, how many biological replicates (n=?) were used to make the graphs. There are no significance tests anywhere in the paper. It should be added where it is appropriate.

Fig 8. In panel A, the 6-well pictures can be removed because they do not add anything to the figure. More accurate quantification is required than counting GFP positive cells in three different sections of the cell cultures. I recommend that they use qPCR for measuring viral DNA load in infected cells. Also, the question is whether virus production was reduced or the infectivity of the virus produced from B(+)Mut was decreased. This result seems to contradict their previous results showing that S13 mutation in B(+) virus (SNU719) increases lytic gene expression.

Fig 5D and 9A-B. I recommend that they quantify their 3C-PCRs using qPCR including multiple biological replicates instead of uisng DNA agarose gels, which often show many different bands questioning specificity of the PCRs.

Reviewer #3: The authors should carefully delineate how many biological replicates were included in each experiment, and note, where appropriate, the statistical significance as determined by the methods stated in their methods section. Without these informations, it is very difficult to assess the rigor, reproducibility and validity of the data presented.

Are the western blots representative from 3 independent experiments?

**Part III – Minor Issues: Editorial and Data Presentation Modifications**

Reviewer #1: The manuscript needs extensive editing to remove ambiguities and English language errors.

Reviewer #2: (No Response)

Reviewer #3: The manuscript is well written, but there are areas of text in the results that are pretty sparse in detail. This makes it difficult for a broad audience to grasp the depth of the experimental findings. For example,

Figure 2- This series of experiments, and what the output data means is confusing. Perhaps it is my unfamiliarity with the specifics of the experiment, but the authors should consider adding text to this section to carefully explain what the MST assay is, what it detects and how this is relevant to the results presented.

other suggestions:

Figure 5. 4C-plots. These are not easy to interpret, especially if the reader is unfamiliar with the EBV genome. The long-range interactions shown are fine, but it would be helpful if a) the EBV genome map was legible on the outside of the Circos plot and b) if the authors helped to identify areas that they specifically discuss in the text (with an arrow maybe on the figure?).

Figure 1: how many reps were done for ChIP seq?

Figure 3: Stats? Also, how were the ChIP assays validated? IgG? another set of cellular controls?

Fig 4: Stats? Replicates?

Fig 7- same comments as figure 3 (see above)

PLOS authors have the option to publish the peer review history of their article (what does this mean?). If published, this will include your full peer review and any attached files.

Reviewer #1: No

Reviewer #2: No

Reviewer #3: No
---

## [Decision Letter · Decision Letter 1]

10 Oct 2022

Dear Dr. Kang,

Thank you very much for submitting your manuscript "Characterization of a new CCCTC-binding factor binding site as a dual regulator of Epstein-Barr virus latent infection" for consideration at PLOS Pathogens. As with all papers reviewed by the journal, your manuscript was reviewed by members of the editorial board and by several independent reviewers. The reviewers appreciated the attention to an important topic. Based on the reviews, we may accept this manuscript for publication, providing that you modify the manuscript according to the review recommendations.

The revised manuscript is much improved, but there remain critical concerns, both in terms of writing as well as specific experiments, that affect rigor and reproducibility. Each experiment must be rigorous even if it is meant to only support other experiments. Please more thoroughly address the remaining reviewer concerns. All of those comments should be addressed, but I would editorially advise in particular, though not exclusive attention to: 1) removal of any experiments without adequate controls and any experiments that are not convincing (the Southern blot appears both not reproducible and perhaps incorrectly interpreted; the 3C assays are "variable" and have no negative control), 2) more clear explanation of how sequencing data confirms deletion of the EcoRI C fragment in B95-8 cells, 3) more clear organization of the MST data including addition of a Coomassie gel of purified CTCF (not just the overexpressed Sf9 lysate). There is a lot of data presented in the revision, but for the next version please emphasize quality over quantity to facilitate a final decision from the journal.

Sincerely,

JJ Miranda

Guest Editor

PLOS Pathogens

Erik Flemington

Section Editor

PLOS Pathogens

Kasturi Haldar

Editor-in-Chief

PLOS Pathogens

orcid.org/0000-0001-5065-158X

Michael Malim

Editor-in-Chief

PLOS Pathogens

orcid.org/0000-0002-7699-2064

The revised manuscript is much improved, but there remain critical concerns, both in terms of writing as well as specific experiments, that affect rigor and reproducibility. Each experiment must be rigorous even if it is meant to only support other experiments. Please more thoroughly address the remaining reviewer concerns. All of those comments should be addressed, but I would editorially advise particular, though not exclusive attention to: 1) removal of any experiments without adequate controls and any experiments that are not convincing (the Southern blot appears both not reproducible and perhaps incorrectly interpreted; the 3C assays are "variable" and have no negative control), 2) more clear explanation of how sequencing data confirms deletion of the EcoRI C fragment in B95-8 cells, 3) more clear organization of the MST data including addition of a Coomassie gel of purified CTCF (not just the overexpressed Sf9 lysate). There is a lot of data presented in the revision, but for the next version please emphasize quality over quantity to facilitate a final decision from the journal.

Reviewer Comments (if any, and for reference):

Reviewer's Responses to Questions

**Part I - Summary**

Reviewer #1: As I have previously reviewed this manuscript, i have focused on the comments I originally made. Unfortunately there are several instances where the authors have not adequately addressed my concerns. These are detailed below:

**Part II – Major Issues: Key Experiments Required for Acceptance**

Reviewer #1: Comment 1: The authors make several comparisons between B95-8 and SNU719 that are directly attributed to the deletion reported to be in B95-8 e.g. Lines 184-186: “Considering that the 11.8-kb EcoRI C fragment (EBV 139724~151554) exists only in the SNU719 EBV genome, it may play an unfavorable role in recruiting CTCF and RNAP II in the SNU719 EBV genome”. How can these comparisons be substantiated if the EcoRI C fragment deletion appears to be in both cell lines? If they cannot resolve the sequencing data to show that the EcoRI C fragment exists in SNU719 and not B95-8, then such comparisons cannot be drawn.

Fig 2A and S6b: A molecular weight marker is still not indicated on the western blots shown in Fig 2A. Line 149 “CTCF protein was of good quality and suitable for subsequent experiments.” By what criteria? The Coomassie stained gel now included (fig S6b) shows the protein is highly contaminated. In fact, compared to the Sf9 lysate there really is no further purification of CTCF protein. The use of such impure protein preparations in sensitive in vitro assays such as MST can give spurious and misleading results. This is demonstrated by the small difference in calculated Kd between wtS13 + CTCF and mtS13 + Sf9, which presumably is not significant judging by the reported standard deviations. Kd values should be reported for all four experimental conditions.

Line 294: still references Fig S4B – should be Fig S4G

Comment 5-1 – has not been fully addressed – “The 3C experiments shown in Fig5D and Figure 9 would benefit from a figure 196 explaining the method and location of primers used.”

Comment-5-2: There do not appear to be any positive or negative controls for this experiment (now fig 5F and G)

Response-5-2: We used “DNA link between 3-kb and 167-kb” as “positive control” in this 3C-PCR 202 assay. “DNA link between 3-kb and 167-kb” was reported in previous studies (reference: attached 203 below). But we did not include any special negative control. 3C-PCR assay showed variable data in our hands. Actually, we observed slightly different results in each 3C-PCR assay and not such convincing these 3C-PCR data.

This comment has not been adequately addressed – the experiment needs a robust negative control. The fact that the authors have declared the variability in their experiment makes me doubt the results further.

Comment 6-2: Now referring to lines 341-344. The densitometry shown is really difficult to interpret. The analysis is outside of the linear range of detection (indicated by maximum intensity being reached in almost all of the bands in the second repeat). It also appears that the authors have cherry picked the blots they analysed – 3/5 blots were excluded because they obtained the ‘best result’ from southern blots 4 and 5. They should either remove these data or provide better analysis.

Comment 7: so this is an n=2 experiment. Statistical analysis should be performed as such otherwise you are including the error in the PCR step, not the ChIP experiment per se.

**Part III – Minor Issues: Editorial and Data Presentation Modifications**

Reviewer #1: (No Response)

PLOS authors have the option to publish the peer review history of their article (what does this mean?). If published, this will include your full peer review and any attached files.

Reviewer #1: No

Figure Files:

Data Requirements:

Reproducibility:

References:

---

## [Editor Report · Decision Letter 2]

15 Dec 2022

Dear Dr. Kang,

We are pleased to inform you that your manuscript 'Characterization of a new CCCTC-binding factor binding site as a dual regulator of Epstein-Barr virus latent infection' has been provisionally accepted for publication in PLOS Pathogens.

Best regards,

JJ Miranda

Guest Editor

PLOS Pathogens

Erik Flemington

Section Editor

PLOS Pathogens

Kasturi Haldar

Editor-in-Chief

PLOS Pathogens

orcid.org/0000-0001-5065-158X

Michael Malim

Editor-in-Chief

PLOS Pathogens

orcid.org/0000-0002-7699-2064

Thank you for your thorough molecular analysis of a new mechanism by which EBV uses three-dimensional genome structure to regulate transcription important for infectivity.
---

## [Editor Report · Acceptance letter]

9 Jan 2023

Dear Dr. Kang,

We are delighted to inform you that your manuscript, "Characterization of a new CCCTC-binding factor binding site as a dual regulator of Epstein-Barr virus latent infection," has been formally accepted for publication in PLOS Pathogens.

Best regards,

Kasturi Haldar

Editor-in-Chief

PLOS Pathogens

orcid.org/0000-0001-5065-158X

Michael Malim

Editor-in-Chief

PLOS Pathogens

orcid.org/0000-0002-7699-2064